# Conformational landscape of HIV-1 Env from closed to fully open

Jiayan Cui[1,2], Zi Jie Lin ®[1,3,5], Sukanya Ghosh[1,5], Jianqiu Du ®[1,4,5], Roopak Sadeesh[1], David B. Weiner ®[1] & Jesper Pallesen ®[1] ✉

The molecular mechanism of HIV-1 entry into host cells is governed by dynamic conformational changes to its envelope glycoprotein (Env), which are triggered by the engagement of the host receptor CD4 and coreceptors. Structural insights into these transitions have been advanced by cryo-electron tomography (cryo-ET), resolving Env structures in closed and multifarious open states within native membranes, and by cryo-electron microscopy (cryo-EM), which has provided atomic details of these states. In this study, we determine cryo-EM structures of soluble native-like Env in complex with antibody 3BC315, antibody b12, CD4, or a combination of 3BC315 and b12, capturing previously uncharacterized conformational states. Observing enhanced 3BC315 binding occupancy in the presence of b12, we investigate the cooperativity of these antibodies using mass photometry and neutralization assays. Integrating these states with the literature, we establish a classification framework for symmetric and asymmetric Env states, categorizing by their degree of openness and stepwise structural rearrangements. Our findings refine the mechanistic understanding of HIV-1 Env dynamics and provide a structural roadmap for targeting dynamic Env states to develop more potent vaccines and immunotherapies.

The HIV-1 envelope glycoprotein trimer (Env) is highly dynamic and essential for viral entry into host CD4$^+$ T cells[1,2]. Env is composed of three gp120 receptor binding subunits and three gp41 transmembrane subunits[2]. The fusion of viral and host membranes is initiated by receptor engagement of gp120 with human CD4, which induces conformational changes in Env that expose binding sites for coreceptors CCR5 or CXCR4[3,4]. Subsequent binding of these human coreceptors triggers insertion of the viral fusion peptide (FP) into the host membrane, eventually leading to viral genome entry, viral replication, and integration[5]. Due to the intrinsic instability of the HIV-1 Env, the SOSIP.664 modification was developed to stabilize soluble Env in its native-like state[6], enabling structural characterization of Env in both closed and CD4-bound open states[7–12]. Comparative analyses of these states have revealed that CD4 binding triggers a cascade of structural rearrangements: burial of FP in a CD4-induced Env pocket, outward

rotation of gp120, formation of four-stranded bridging sheet and α0 helix, and displacement of V1V2 loops to expose the coreceptor binding site[10–12].

Recently, cryo-EM and cryo-ET studies have elucidated stepwise CD4 engagement with Env, providing biological relevance to the structural intermediates of SOSIP-stabilized Env[13,14]. CD4 engagement begins with the binding of one CD4 molecule, resulting in either a closed or an asymmetric open conformational state of Env. Subsequent binding of a second or even a third CD4 molecule results in the asymmetric or symmetric opening of Env, respectively, accompanied by a ~50 Å reduction in viral-envelope-to-host-membrane distance. Notably, cryo-ET reveals that binding of three membrane-attached CD4 molecules results in Env adopting a partially open state distinct from the fully open state typically observed by cryo-EM[11,14]. Additionally, occluded open states – exhibiting gp120 rotation but without

[1]Vaccine and Immunotherapy Center, The Wistar Institute, Philadelphia, PA, USA. [2]Department of Bioengineering, University of Pennsylvania, Philadelphia, PA, USA. [3]Department of Chemistry, Indiana University, Bloomington, IN, USA. [4]Department of Molecular and Cellular Biochemistry, Indiana University, Bloomington, IN, USA. [5]These authors contributed equally: Zi Jie Lin, Sukanya Ghosh, Jianqiu Du. ✉e-mail: jpallesen@wistar.org

V1V2 displacement, four-stranded bridging sheet or α0 helix formation – have been captured by antibody binding and characterized by both cryo-ET and cryo-EM[10,15–17]. These occluded open states, alongside evidence that Env spontaneously samples open conformations without CD4[12,18–21], suggest that CD4 engages Env in a pre-existing conformational continuum.

In this study, we characterized distinct conformations of AMC008 Env from a clade B Tier-1B virus to delineate the interplay between closed, occluded open, and CD4-bound open states. AMC008 Env has been reported as highly conformationally heterogeneous; however, SOSIP.v4.2 stabilization (H66R, A316W, I535M, L543N, and SOSIP.664) enables expression of soluble, flexible AMC008 SOSIP.v4.2 Env (hereafter AMC008)[22]. We identified distinct Env conformations by complexing AMC008 with antibody 3BC315 (targeting the gp120-gp41-gp41 interface)[23], antibody b12 (targeting the CD4 binding site; CD4bs)[24], CD4, or a combination of 3BC315 and b12. 3BC315 and b12 were chosen based on prior insights: b12 facilitated the cryo-EM structure of occluded open Env[10], while 3BC315 destabilizes Env via its interactions with the interprotomer interface[25]. Cryo-EM structures of these complexes (resolutions ranging between 2.9–3.9 Å), alongside a closed reference structure determined to a resolution of 2.9 Å, unveiled distinct conformational states including a base-relaxed state upon 3BC315 binding, occluded moderately open states upon b12 or CD4 binding, and an asymmetric occluded open state upon combined b12 and 3BC315 binding. Integrating these results with previous studies on soluble Env, we propose a classification framework to characterize Env conformations and a mechanism by which HIV-1 Env transitions from closed to CD4-bound fully open en route to membrane fusion. Lastly, we demonstrate increased cooperativity between 3BC315 and b12, two neutralizing antibodies from different human subjects, in terms of affinity and neutralization. Our findings refine the mechanistic understanding of HIV-1 Env dynamics and provide a structural roadmap for targeting dynamic Env states in more potent vaccine and immunotherapeutic designs.

## Results

### 3BC315 binds a quaternary epitope

3BC315 binding to Env was previously reported to dissociate Env trimers[25]. To stabilize the AMC008-3BC315 complex, we added VRC01 targeting the CD4bs and PGT121 targeting the V3-glycan epitope[26,27]. We determined a cryo-EM structure of the resulting AMC008-PGT121-VRC01-3BC315 complex at 3.2 Å (Fig. 1a). In the data set, we observed two classes having complexed either one copy of 3BC315 or none at all, whereas both classes exhibit full VRC01 and PGT121 occupancy (Supplementary Fig. 1b). This structure reveals atomic details of the 3BC315 epitope-paratope interface. To assess conformational changes in 3BC315 upon Env engagement, we superimposed the Env-bound 3BC315 Fab with its unliganded crystal structure (Cα RMSD = 0.6 Å; Supplementary Fig. 2b)[25]. While the two structures are highly similar, key differences include the side chain reorientation of $F56_{CDRH2}$ to accommodate the $N88_{gp120}$ glycan and the side chain reorientation of $R27_{CDRL1}$ to hydrogen-bond with $N543_{gp41}$ (Supplementary Fig. 2c).

3BC315 binds predominantly to one gp41 subunit but exhibits additional interactions with gp120 (same protomer) and adjacent gp41. The epitope is bordered by $N88_{gp120}$ and $N625_{gp41}$ glycans (Fig. 1b). The $N88_{gp120}$ glycan is axially flipped 180° and tilted 70° toward the Env apex to accommodate 3BC315 (Fig. 1c). Both 3BC315 heavy and light chains interact extensively with AMC008 with buried surface areas of 915 Å² and 503 Å² on the epitope, respectively (Fig. 1b). CDRH2, CDRH3, CDRL1, and CDRL3 of 3BC315 form polar interactions with the epitope to stabilize the binding of 3BC315 (Fig. 1d). A hydrophobic groove formed by six aromatic residues (Y30 and F32 of CDRL1; Y91, Y94, Y100E, and F100I of CDRL3) accommodates an Env α-helix spanning the fusion peptide proximal region (FPPR) and the N-terminal region of heptad repeat 1 (HR1N) (Fig. 1e). Additionally, the

3BC315 CDRH3 loop inserts into a cavity at the gp120-gp41-gp41 interface, positioning its tip at the hydrophobic gp120-gp41 interface (Fig. 1f).

### 3BC315-bound base-relaxed AMC008

To enable comparison of our 3BC315-bound AMC008 structure with closed AMC008, we solved a 2.9 Å cryo-EM structure of closed AMC008 by complexing AMC008 with VRC01 and 35O22 (targeting the gp120-gp41 interface) (Fig. 2a)[28].

Superimposition of the AMC008-PGT121-VRC01-3BC315 and AMC008-VRC01-35O22 structures demonstrates dramatic displacements by more than 5 Å in epitope-associated structural elements upon 3BC315 binding (Fig. 2b). A hinge region comprising residues $L523_{gp41}$-$S534_{gp41}$ undergoes rigid-body movement, coupled with inward movement of FP toward the Env core and outward movement of HR1N (Fig. 2b, c). Within the 3BC315-bound protomer, FP is embedded in a cavity formed by the non-helical α0 loop, the β0 strand, the α0-β0 connecting loop, and the remodeled HR1N (Fig. 2d). Furthermore, we aligned our 3BC315-bound gp41 to other gp41 remodeled Envs in the literature[10,11,29–32]. The 3BC315-bound gp41 resembles gp41s of T/F100-8ANC195 or CAP256.wk34.c80 SOSIP.RnS2, where, as in our AMC008-PGT121-VRC01-3BC315, the FP burial occurs despite retention of closed gp120 conformation (Fig. 2e, f). Notably, FP burial is a hallmark of Env opening[10]. In occluded partially open CNE40-HmAb64 and CD4-bound partially open BG505-8ANC195-17b-CD4, FPs adopt conformations similar to 3BC315-bound gp41 but are less extended to avoid clashes with the C-terminal region of HR1 (HR1C). In contrast, in occluded fully open B41-b12 and CD4-bound fully open B41-17b-CD4, FP burial persists, but with further retraction to accommodate HR1C (Fig. 2f). Based on this data, we conclude that FP conformation in 3BC315-bound AMC008 gp41 is equivalent to FP conformations in T/F100-8ANC195 and CAP256.wk34.c80 SOSIP.RnS2, and this conformation most resembles partially open states. Thus, we define this FP buried closed conformational state as the base-relaxed state. In contrast, FP of BG505-PGT122-VRC34.01 sits in an opposite orientation away from the Env core directed by antibody VRC34.01 (Fig. 2f). Taken together, these results underscore FP conformational plasticity across closed, partially open, and fully open Env states.

To exclude potential confounding effects of VRC01/PGT121 on our observed base-relaxation, we determined cryo-EM structures of AMC008 bound solely to 3BC315. Unlike the AMC008-PGT121-VRC01-3BC315 complex that exhibited no more than single 3BC315 occupancy, the 3BC315-only dataset resolved two 3D classes: AMC008 bound to one (3.6 Å) or two (3.5 Å) 3BC315 (Supplementary Figs. 2a, 3a). More importantly, all 3BC315-bound protomers adopt the base-relaxed conformation, while unbound protomers retain the closed conformation (Supplementary Fig. 2d), confirming that remodeling is driven by 3BC315 binding rather than auxiliary antibodies.

### b12-bound moderately open AMC008

We next determined the cryo-EM structure of AMC008 in complex with b12 at a resolution of 3.9 Å (Fig. 3a). Surprisingly, the degree of gp120 opening observed in the AMC008-b12 structure is significantly reduced compared to the previously resolved B41-b12 structure[10]. To quantify gp120 openness in AMC008-b12, we measured interprotomer distances for representative Cα residues at the V3 base ($H330_{gp120}$), V1V2 base ($P124_{gp120}$), and CD4bs ($D368_{gp120}$). These measurements were compared across closed (AMC008-VRC01-35O22), partially open (CNE40-HmAb64, BG505-8ANC195-17b-CD4), and fully open (B41-b12, B41-17b-CD4) Env structures (Supplementary Fig. 4a, b, d–g)[10,11,32]. The AMC008-b12 openness lies between closed and partially open states, leading us to define this as a moderately open state. Subsequent alignment of gp41 across these structures revealed distinct HR1C conformations: in partially and fully open states, the HR1C helix extends by at least one additional N-terminal turn and rotates in

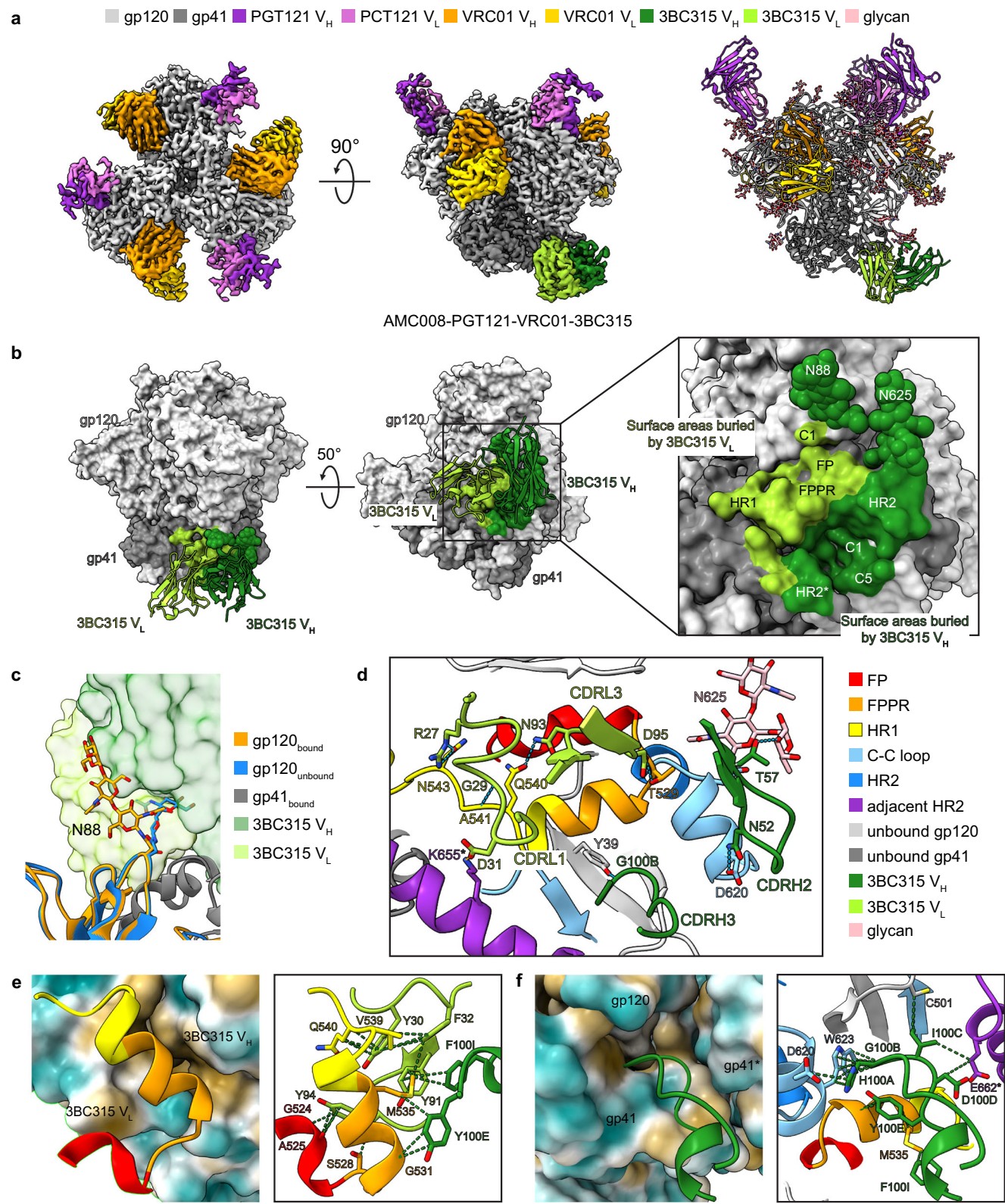

multiple directions relative to the closed state. While AMC008-b12 also exhibits N-terminal helical extension of HR1C, its orientation remains similar to a closed state (Fig. 3e). The apparent movement of HR1C arises from rigid-body shifts relative to the C-C loop/HR2 module, which is used for alignment, and not from internal rearrangements within HR1C itself. Integrating gp120 openness, gp41 conformation, and CD4-binding status, we categorize Env states as follows: A (closed), A_BR (base-relaxed closed), B (occluded moderately open), C (occluded

partially open), C' (CD4-bound partially open), D (occluded fully open), and D' (CD4-bound fully open).

Notably, significant displacement was observed at the β4̄/β26 strands of the gp120 N- and C-termini, which insert into the gp41 'four-helix collar' – identifying this region as a pivotal point for gp120 rotation (Fig. 3b). Displacements in the V1, V2, and V4 loops likely reflect inherent loop flexibility and isolate-specific sequence differences. Docking b12 Fab onto closed AMC008 reveals steric clashes

**Fig. 1 | Epitope analysis of 3BC315 in AMC008 SOSIP.v4.2-PGT121-VRC01-3BC315. a** 3.2 Å cryo-EM density map (left and middle, top and side views) and atomic model (right) of AMC008 in complex with PGT121, VRC01, and 3BC315 Fabs. **b** Left and middle: surface areas of AMC008 buried by 3BC315 Fab. Right: close-up view of the 3BC315 binding epitope. **c** Glycan N88 is flipped to accommodate 3BC315. **d** Hydrophilic interactions at the AMC008-3BC315 epitope-paratope interface. **e** Left: G524-Q540 of AMC008 interacts with a highly hydrophobic groove at the interface of 3BC315 heavy chain and light chain formed by six aromatic amino acid residues. The surfaces of 3BC315 are colored by hydrophobicity. Right: hydrophobic contacts at the interface. **f** Left: 3BC315 CDRH3 is inserted into a hydrophobic hole on the AMC008 gp120-gp41-gp41* interface. The surfaces of AMC008 are colored by hydrophobicity. Right: hydrophobic contacts at the interface. Hydrogen bonds are indicated by blue dashed lines; salt bridges are indicated by orange dashed lines; hydrophobic contacts are indicated by green dashed lines. Amino acids from the adjacent protomer are labeled with * marks. The coloring schemes for the ribbon representation in **e** and **f** are the same as in **d**. PGT121 and VRC01 Fabs are hidden in **b** for better visualization. $V_H$, heavy chain variable domain; $V_L$, light chain variable domain; FP, fusion peptide; FPPR, fusion peptide proximal region; C-C loop, disulfide loop; HR1, heptad repeat 1; HR2, heptad repeat 2.

with N197 and N301 glycans (Fig. 3c). Upon b12 binding, the N197 glycan reorients to avoid clashes, accompanied by limited gp120 opening. This limited opening is sufficient for B12 accommodation, suggesting that SOSIP.v4.2-stabilized AMC008 samples open states less readily than B41 SOSIP (Fig. 3d).

The gp41 conformation in AMC008-b12 (state B) resembles gp41 of the closed state A, with solvent-exposed FP and bent FPPR/HR1N helix. However, HR1C in AMC008-b12 exhibits a 4.4° axial rotation toward the central symmetry axis, and HR1N in AMC008-b12 is better resolved compared to state A (Fig. 3f). Trimeric gp41 assemblies align closely between CD4-bound and unbound states for both partially open (C/C′) and fully open (D/D′) conformations (Fig. 3g, h). Notably, while HR1C conformations in states B, C/C′, and D/D′ appear distinct when aligned by protomer, their overall helical bundle compactness is similar when aligning full trimeric assemblies (Fig. 3i).

## CD4-bound moderately open AMC008

Observing distinct conformational differences between occluded open AMC008-b12 and B41-b12 structures, we hypothesized that CD4 binding to AMC008 would induce an alternative conformation. We therefore determined the cryo-EM structure of the AMC008-CD4 complex at a resolution of 2.9 Å (Fig. 4a). Interprotomer distance measurements revealed that CD4-bound AMC008 exhibits intermediate gp120 openness, positioned between closed and partially open Envs but less open than AMC008-b12 (Supplementary Fig. 4a–g). CD4 binding destabilizes interprotomer contacts at the trimer apex, rendering the V1V2V3 region more flexible and disordered. Local classification of the apex (Supplementary Fig. 5a) yielded a 3.6 Å map showing that AMC008-CD4 gp120 retains occluded Env-like features: disordered but non-displaced V1V2 loops maintain the coreceptor binding site occluded despite partly disordering of V3, with no formation of the α0 helix or 4-stranded bridging sheet (Fig. 4b). This structural arrangement explains the finding that CD4 binding to AMC008 SOSIP.v4.2 only marginally enhances binding of 17b, an antibody targeting the coreceptor binding site[22]. Additionally, two previous reports identified single CD4-bound Envs in closed conformation with occluded coreceptor binding sites (state A′)[13,33]. As AMC008-CD4 retains gp41 assembly similarity to AMC008-b12 despite differing gp120 openness, we classify it as state B′ (CD4-bound occluded moderately open) under our established criteria (Fig. 4c).

The binding of CD4 does not result in the formation of a 4-stranded bridging sheet in AMC008-CD4, however, the coordinates of β20/β21 align well with other CD4-bound Env gp120s to form the essential contacts with the CD4 F43 residue, indicating the remodeling of β20/β21 is presumably the first step of CD4 engagement (Fig. 4d). This aligns with prior studies showing that restricting β20/β21 mobility via an interdomain C113-C429 disulfide bond impairs CD4 binding, underscoring the importance of β20/β21 remodeling for CD4 engagement[34]. Comparisons of the closed, the CD4-bound, and the b12-bound AMC008 structures revealed that $W571_{gp41}$ transitions from deeply buried between α0-1 and β4̄ (state A/A′) to exiting this hydrophobic pocket in states B/B′ as a consequence of HR1C helix bundle compaction (Fig. 4e).

Based on this analysis, we find the HIV-1 Env trimer adopts distinct conformational states characterized by progressive gp120 opening and gp41 rearrangements. The closed state A features tightly packed gp120 subunits at the trimer apex, solvent-exposed FP, bent FPPR/HR1N helices, and a non-extended HR1C bundle. A variant of this state, base-relaxed closed state $A_{BR}$, retains most features of state A but adopts a gp41 base conformation resembling the partially open state C. In the occluded moderately open state B, gp120s rotate outward moderately, while gp41 retains state A-like features except for an extended and compacted HR1C bundle. Further opening defines the occluded partially open state C, where gp120s rotate more prominently, FP buries as a helical segment, and FPPR/HR1N adopts bent or straight conformations. In the occluded fully open state D, gp120 rotates to its maximal extent, FP is repositioned into a shorter loop to avoid clashes, and FP/FPPR adopts a straight helix conformation. CD4 binding induces state-specific changes: it leaves state A′ unaltered relative to state A, remodels β20/β21 strands in state B′, displaces V1V2 while forming the α0 helix and the 4-stranded bridging sheet to expose the coreceptor binding site in states C′/D′ (Table 1).

## 3BC315 and b12-bound asymmetric open AMC008

Having observed alternative conformational changes upon binding either b12 or 3BC315, we rationalized that complexing AMC008 with both antibodies might lead to further structural rearrangements. We determined a cryo-EM structure of the AMC008-b12-3BC315 complex at a resolution of 3.7 Å (Fig. 5a). Unexpectedly, with the presence of b12, the binding occupancy of 3BC315 increases from one or two in our base-relaxed structures to three. Additionally, unlike the limited degree of gp120 opening observed in the AMC008-b12 structure, gp120s of the AMC008-b12-3BC315 structure are more open but in an asymmetric manner. Heavy chain framework region (FR1) of one b12 Fab interacts with light chain $C_L$ domain of one 3BC315 Fab. 3D variability analysis of the AMC008-b12-3BC315 density map confirmed these interactions, revealing that one b12/3BC315 Fab pair interaction is stable, the other two engage in transient contacts, indicating a cooperative binding mechanism (Supplementary Movie 1).

Despite the global conformational opening of AMC008 and increased 3BC315 binding stoichiometry compared to our solely 3BC315-bound structures, there are only subtle local conformational changes in FP C-terminus and FPPR of the 3BC315 epitope. However, HR1C in each protomer rotates, becomes more compact, and forms additional helical turns, forcing FP to adopt a less extended conformation to prevent clashes with the rotated HR1C compared to the base-relaxed $A_{BR}$ state (Fig. 5b).

We next compared AMC008-b12-3BC315 to other published asymmetric open Env structures (Supplementary Fig. 4h–s)[12,13,17,35–37]. This gp120 conformation of AMC008-b12-3BC315 is similar to two published CD4-bound asymmetric open Envs, HT2-CD4 and BG505-E51-CD4 class II, evaluated by measuring the interprotomer residue distances of gp120 in these structures and global alignments (Fig. 5c and Supplementary Fig. 4h–j). Alignment of AMC008-b12-3BC315 to HT2-CD4 and BG505-E51-CD4 class II by protomer revealed that they

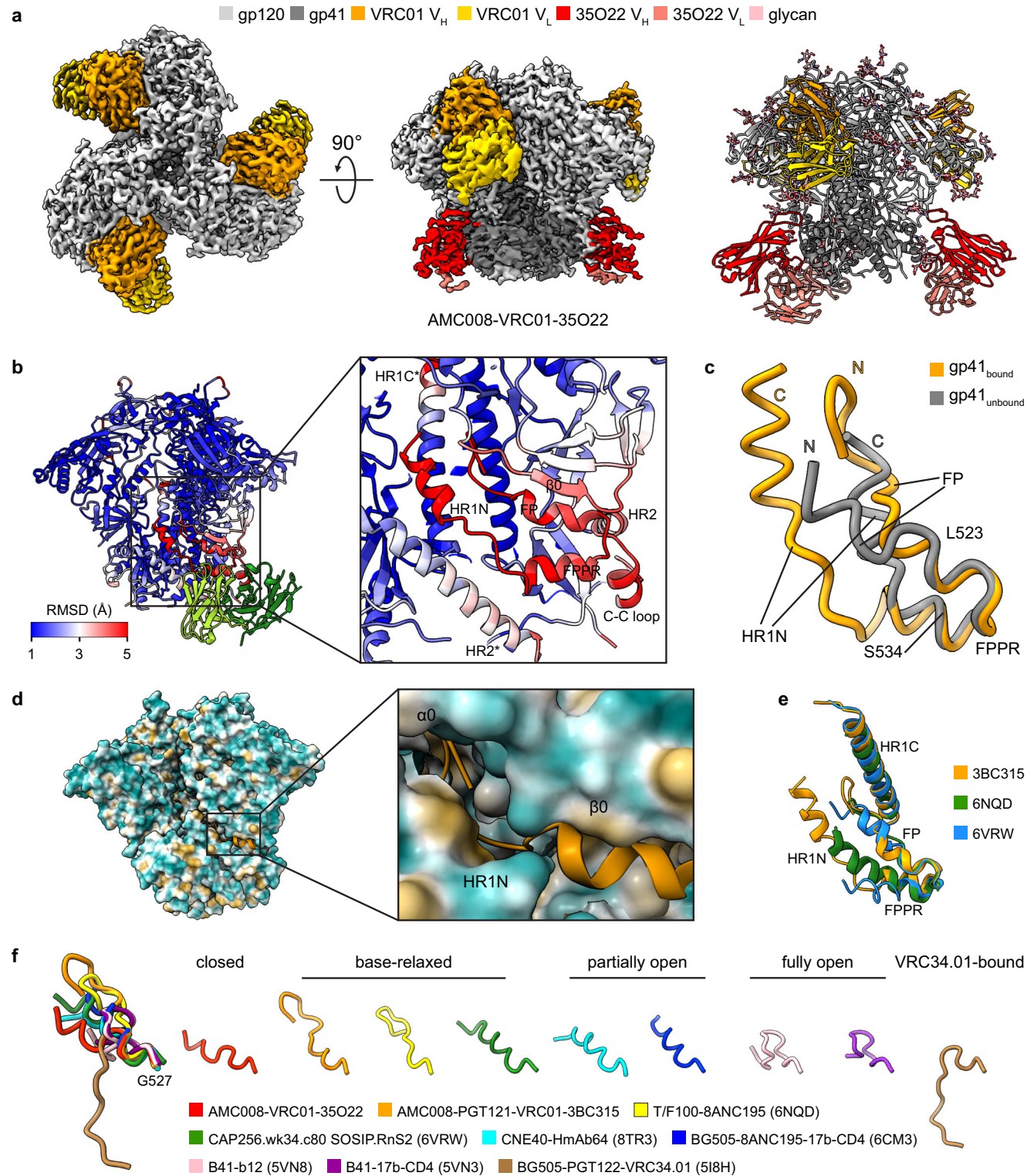

**Fig. 2 | Remodeling of AMC008 SOSIP.v4.2 upon 3BC315 binding. a** 2.9 Å cryo-EM density map (left and middle, top and side views) and atomic model (right) of AMC008 in complex with VRC01 and 35O22 Fabs. **b** Left: Cα RMSD of 3BC315-bound AMC008 in comparison to closed AMC008. Right: a close-up view of the 3BC315 binding epitope. **c** Comparison of FP, FPPR, and HR1N conformations with and without 3BC315 binding. C denotes C-terminus, and N denotes N-terminus. **d** Left: conformation of FP upon 3BC315 binding with surface representation of AMC008. Right: a close-up view shows FP is embedded in the cavity formed by α0, β0, and HR1N. The surfaces of AMC008 are colored by hydrophobicity.

**e** Comparison of gp41 conformations in AMC008-PGT121-VRC01-3BC315, T/F100-8ANC195 (6NQD [https://doi.org/10.2210/pdb6NQD/pdb]), and CAP256.wk34.c80 SOSIP.RnS2 (6VRW [https://doi.org/10.2210/pdb6VRW/pdb]) Env structures. **f** Comparison of FP conformations in Env structures. PGT121 and VRC01 Fabs are hidden in (**b**), PGT121, VRC01, and 3BC315 Fabs are hidden in (**d**) for better visualization. V_H heavy chain variable domain, V_L light chain variable domain, FP fusion peptide, FPPR fusion peptide proximal region, HR1N N-terminal region of heptad repeat 1, HR1C C-terminal region of heptad repeat 1.

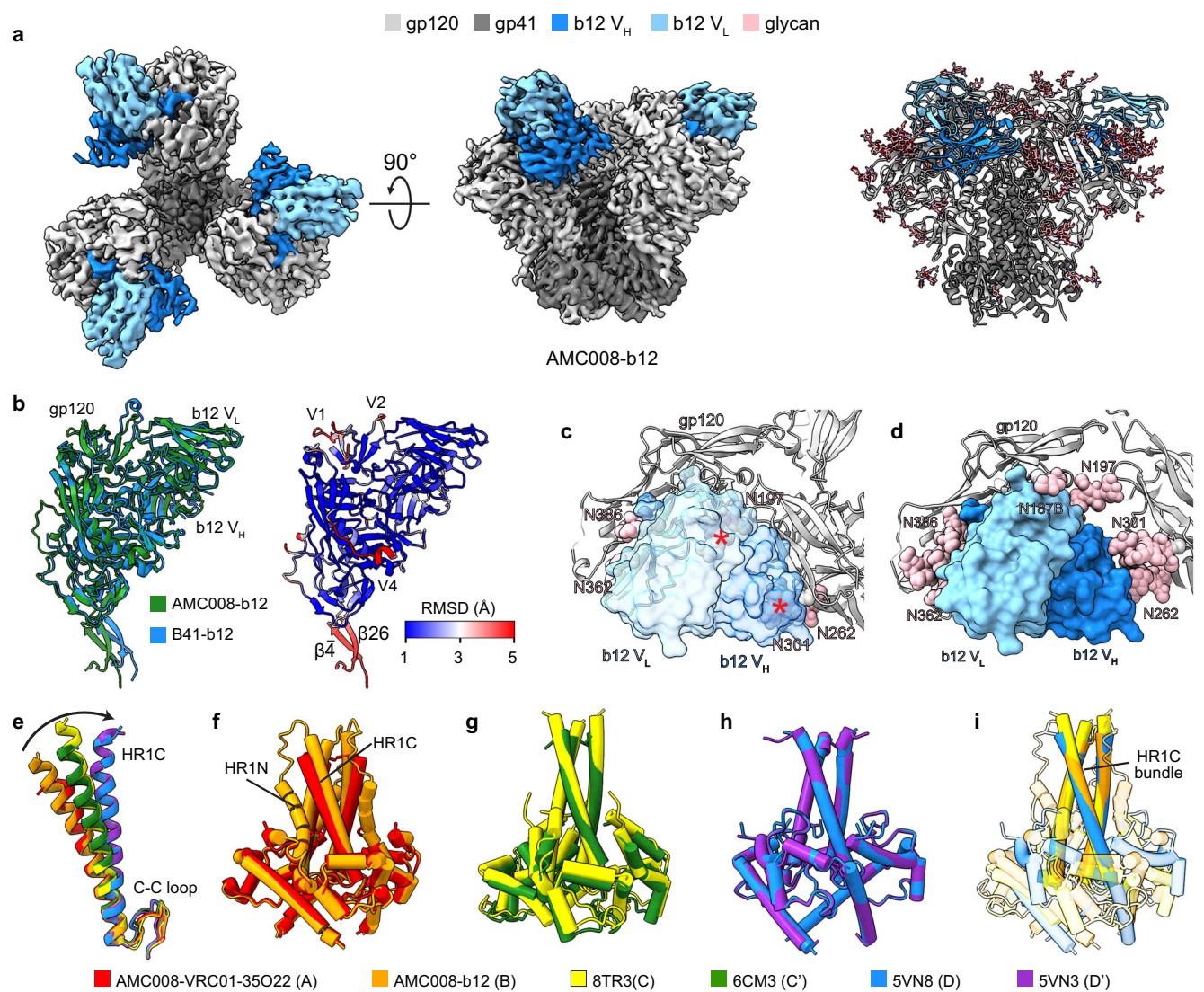

**Fig. 3 | Remodeling of AMC008 SOSIP.v4.2 upon b12 binding. a** 3.9 Å cryo-EM density map (left and middle, top and side views) and atomic model (right) of AMC008 in complex with b12 Fab. **b** Left: superimposition of the protomers of b12-bound B41 and AMC008. Right: Cα RMSD of b12-bound AMC008 protomer in comparison to b12-bound B41 protomer. **c** b12 cannot bind AMC008 in closed conformation due to clashes of N197 and N301 glycans with b12. Clashes are indicated by red * marks. **d** Binding of b12 requires minimum openness of AMC008 gp120 to accommodate N197 and N301 glycans. **e** Comparison of the HR1C conformation of AMC008-b12 with symmetric closed, partially open, and fully open

Envs. **f–h** Superimposition of gp41s in AMC008-VRC01-35O22 and AMC008-b12 (**f**), CNE40-HmAb64 (8TR3 [https://doi.org/10.2210/pdb8TR3/pdb]) and BG505-8ANC195-17b-CD4 (6CM3 [https://doi.org/10.2210/pdb6CM3/pdb]) (**g**), B41-b12 (5VN8 [https://doi.org/10.2210/pdb5VN8/pdb]) and B41-17b-CD4 (5VN3 [https://doi.org/10.2210/pdb5VN3/pdb]) (**h**). **i** Superimposition of HR1Cs in AMC008-b12, CNE40-HmAb64, and B41-b12. A, B, C, and D denote closed, moderately open, partially open, and fully open states. CD4-bound states are denoted with '. V_H heavy chain variable domain, V_L light chain variable domain, HR1N N-terminal region of heptad repeat 1, HR1C C-terminal region of heptad repeat 1, C-C loop disulfide loop.

are similar at each position regardless of CD4 binding (Fig. 5d). Despite these similarities, gp120s of AMC008-b12-3BC315 lack the features of a CD4-bound open conformation, which are displaced V1V2 loops, accessible coreceptor binding site, formation of α0 helix and 4-stranded bridging sheet. We further compared gp41 conformations of AMC008-b12-3BC315 by aligning each protomer to the symmetric open reference structures (Fig. 5e). We observed that gp41 of each protomer can be matched with states C, or D, specifically, protomer 1 aligns well with state C while protomers 2 and 3 align better with state D. Therefore, we denote the b12 and 3BC315 bound AMC008 as state CDD. Furthermore, protomers of the currently existing asymmetric open Envs can be categorized using the same approach; thus, our state classification framework for symmetric Envs can be branched out to properly denote asymmetric Envs (Fig. 5e and Supplementary Table 1).

From these structures of asymmetric or symmetric Envs, we further differentiated protomeric states C/C' and D/D' based on the coordination of W571$_{gp41}$: the side chain of W571$_{gp41}$ is above the α0-β0 connecting loop in state C/C'; but is beneath the loop and is buried in a hydrophobic pocket formed by F53$_{gp120}$, P79$_{gp120}$, C218$_{gp120}$, and C247$_{gp120}$ in state D/D' (Fig. 5f). Together with states A/B/B', we identify W571$_{gp41}$ to be a crucial residue that changes position corresponding to the conformational state of HIV-1 Env, consistent with previous structural and functional studies of Env W571$_{gp41}$[12,38,39].

To study the biological relevance of the states C' and D', we docked Env models of states C'/D' into a cryo-ET density map of three CD4-bound Env, and then aligned gp120-CD4-CCR5 and AlphaFold-predicted full-length CD4 models to the docked Envs (Fig. 5g)[14,40,41]. Using three copies of full-length CD4 and gp120-CD4-CCR5, we

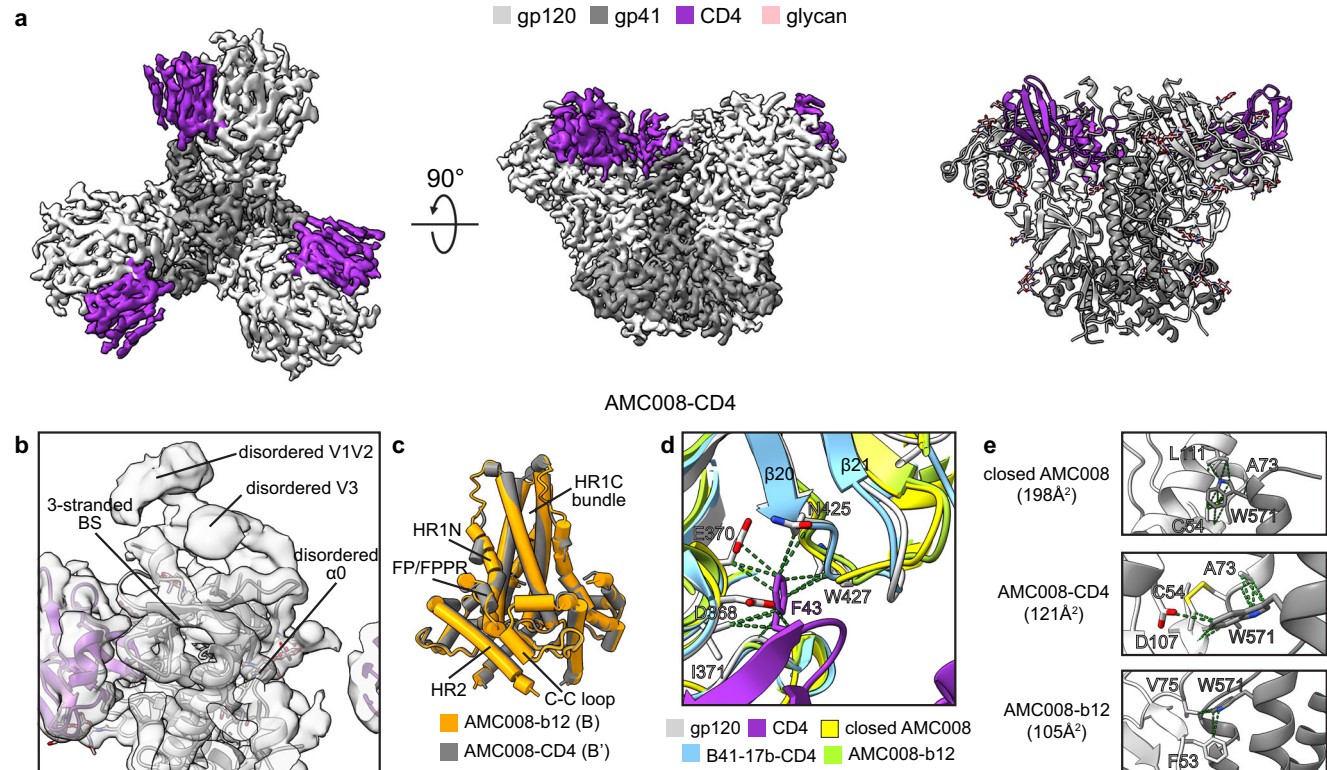

AMC008-CD4

**Fig. 4 | Remodeling of AMC008 SOSIP.v4.2 upon CD4 binding. a** 2.9 Å cryo-EM density map (left and middle, top and side views) and atomic model (right) of AMC008 in complex with CD4 D1D2. **b** AMC008-CD4 model fitted in a 3.6 Å unsharpened map. **c** Superimposition of gp41s of AMC008-b12 and AMC008-CD4. **d** Superimposition of AMC008-CD4 with AMC008-VRC01-35O22, B41-17b-CD4 (5VN3), and AMC008-b12 shows remodeling of β20/β21 upon binding of CD4.

Green dashed lines indicate contacts between the CD4 F43 residue and AMC008 gp120. **e** The W571$_{gp41}$ residue becomes less buried upon binding of b12 or CD4. BS bridging sheet, FP fusion peptide, FPPR fusion peptide proximal region, C-C loop disulfide loop, HR1N N-terminal region of heptad repeat 1, HR1C C-terminal region of heptad repeat 1, HR2 heptad repeat 2.

approximated the membrane surfaces of CCR5 and CD4. The approximated membrane surface of CD4 in state C′ matches the membrane density, while state D′ does not, consistent with the observation that Env in complex with CD4 imaged by cryo-ET adopts a partially open state[14]. We discovered that the membrane surfaces for CCR5 and CD4 do not positionally match in state C′ but appear to be much closer in D′. Based on this analysis, we predict that the virus-to-host membrane distance for state D′ will likely be ~110 Å, while the distance is measured to be ~140 Å in state C′ by cryo-ET[14]. Therefore, we expect states A′, B′, and C′ all to be early states that do not engage the co-receptor, while state D′ is likely primed to engage with the coreceptor and followed by membrane insertion of FP and dissociation of gp120.

**Structural and functional cooperativity of b12 and 3BC315**

Cryo-EM analysis of the AMC008-3BC315 complex revealed two distinct populations: 48.2% of particles are bound to a single 3BC315 Fab, while 51.8% accommodate two Fabs per trimer (Supplementary Fig. 3a). Interestingly, in the AMC008-PGT121-VRC01-3BC315 complex, 67.4% of particles retain one 3BC315 Fab, with the remaining 32.6% lack 3BC315 entirely (Supplementary Fig. 1b). In contrast, the AMC008-b12-3BC315 complex exhibits uniform occupancy, with three 3BC315 Fabs bound per trimer. These findings suggest that VRC01 and PGT121 stabilize Env in the closed state (State A), which allosterically impedes 3BC315 binding, whereas b12 promotes a more open state (State B), enabling full 3BC315 occupancy.

To validate the cooperative binding of b12 and 3BC315 to Env, we performed mass photometry experiments (Table 2 and Supplementary Fig. 6). The molecular mass of trimeric native-like AMC008 was estimated at approximately 338 kDa. Upon binding excess 3BC315 Fab,

the mass increased by 114 kDa (consistent with ~2 Fabs), whereas excess b12 Fab or VRC01 Fab increased the mass by 170 kDa or 164 kDa, respectively (~3 Fabs each). Consistent with our cryo-EM results, adding excess 3BC315 Fab to the AMC008-b12 complex induced an additional 162 kDa increase (~3 Fabs). A distinct peak corresponding to AMC008-b12 without 3BC315 binding was observed, suggesting an all-or-none binding stoichiometry for 3BC315 Fab in the presence of b12. This implies that, in the context of b12, initial 3BC315 Fab binding markedly enhances affinity for the remaining two sites, indicative of positive cooperativity. In contrast, adding excess 3BC315 Fab to the AMC008-VRC01 complex yielded only a 66 kDa increase (~1 Fab), reflecting negative cooperativity. These results align with our cryo-EM data, confirming that b12 facilitates 3BC315 binding to the AMC008 trimer, while VRC01 antagonizes it.

b12 is a CD4bs bNAb that exhibits limited binding and neutralization to tier 2 or higher HIV isolates, likely due to restricted conformational flexibility in these Env trimers. Our structures of AMC008-b12 and AMC008-b12-3BC315 reveal that while b12 binds to moderately open Env, the combination of both b12 and 3BC315 leads to an even more open Env conformation. We hypothesized that this cooperative binding, observed structurally, would enhance neutralization compared to either antibody alone, even against resistant isolates. To test this, we performed pseudovirus neutralization assays. 3BC315 neutralized all tested strains, whereas b12 showed activity only against SF162 and BaL, as previously reported and confirmed by our data (Fig. 6a, b)[23,24]. When the two antibodies were combined (Fig. 6c–g), our neutralization assays showed a synergistic increase in neutralization activity, suggesting enhanced engagement of Env on the viral envelope. This neutralization synergy was pronounced for strains that could be neutralized by either b12 or 3BC315 – SF162 and BaL

**Table 1 | Summary of structural features of Env conformational states with or without CD4 binding**

|  | gp120 | V1V2 | αO | CoRBS | FP | BS | HR1C | HR1N&FPPR |
|---|---|---|---|---|---|---|---|---|
| State A | closed | contacting | disordered/loop | occluded | solvent exposed loop | 3-stranded | separated | bent helix |
| State A' | closed | contacting | disordered/loop | occluded | solvent exposed loop | 3-stranded | separated | bent helix |
| State A$_{BR}$ | closed | contacting | disordered/loop | occluded | buried loop | 3-stranded | separated | bent helix |
| State B | moderately open | not displaced | disordered/loop | occluded | solvent exposed loop | 3-stranded | extended, compact | bent helix |
| State B' | moderately open | disordered | disordered/loop | occluded | solvent exposed loop | 3-stranded | extended, compact | bent helix |
| State C | partially open | not displaced | disordered/loop | occluded | buried helix | 3-stranded | extended, compact | bent helix/helix |
| State C' | partially open | displaced | helix | accessible | buried helix | 4-stranded | extended, compact | bent helix/helix |
| State D | fully open | not displaced | disordered/loop | occluded | buried loop | 3-stranded | extended, compact | helix |
| State D' | fully open | displaced | helix | accessible | buried loop | 4-stranded | extended, compact | helix |

CoRBS coreceptor binding site, FP fusion peptide, BS bridging sheet, HR1C C-terminal region of heptad repeat 1, HR1N N-terminal region of heptad repeat 1, FPPR fusion peptide proximal region.

(Fig. 6d, e). Interestingly, even for strains completely resistant to b12 neutralization – BG505_T332N and TRO11, the presence of b12 still enhanced the neutralization activity when in combination with 3BC315, though less significantly than against strains that can be neutralized by either b12 or 3BC315 (Fig. 6c, f). These findings demonstrate that structural cooperativity between b12 and 3BC315 correlates with functional enhancement, with implications for antibody combination therapies.

## Discussion

In this study, we solved cryo-EM structures of base-relaxed AMC008 with 3BC315 at atomic resolution, revealing conformational changes that resemble open Env conformations at the gp41 binding interface, namely FP, FPPR, and HR1N. Additionally, we solved the structures of AMC008 bound with CD4 or b12 alone and with 3BC315. In our B12-bound or CD4-bound AMC008 structures, we found a distinct Env conformational intermediate, coined moderately open conformation, between the closed and partially open Env conformations with a moderately rotated gp120 and an N-terminally extended and more compacted HR1C. While 3BC315 was found to induce a local gp41 conformational change without changing the conformation of gp120, our structure of b12 and 3BC315 in complex with AMC008 revealed an alternative asymmetric occluded open Env conformation that resembles the two CD4-bound and three CD4-bound asymmetric open Env structures[12,13]. We observe full occupancy (three copies of b12 and three copies of 3BC315 bound) in this asymmetric occluded open state, and hydrogen-bonding ability between b12 FR1 and 3BC315 $C_L$ domain. These structural arrangements suggest a cooperative mechanism between the two antibodies that combines antibody stacking and manipulation of HIV-1 Env allostery to enhance combined antibody affinities to their respective epitopes. We confirmed our structural observations of antibody cooperativity using mass photometry and pseudovirus neutralization assays. Utilizing the effect of cooperativity between a combination of antibodies could be a potential advantage when developing prophylactic or therapeutic antibody cocktails, leading to overall increased potency and breadth. Furthermore, 3BC315 exhibits the distinct property of capturing an alternative gp41 conformation in a closed Env (AAA$_{BR}$) that potentially shifts the conformational equilibrium of HIV-1 Env towards more open and immunogenic states. Developing vaccination strategies to elicit these types of broadly neutralizing antibodies could be another viable prophylactic avenue due to their ability to overcome the increased neutralization susceptibility of Env toward neutralizing antibodies that bind open conformations.

Understanding the conformational flexibility of HIV-1 Env has long been challenging, in part due to the absence of a systematic framework for classifying intermediates along the closed-to-open continuum, particularly for pre-CD4-engaged states. Recent cryo-EM and cryo-ET studies have revealed that asymmetric Env conformations are biologically relevant, occurring in trimers bound with one, two, or three CD4 molecules[13,14]. Complementary smFRET analyses demonstrate that even unliganded Env on native virions spontaneously samples similar conformational states, which become further enriched following CD4 engagement[19,42,43]. Additionally, biophysical techniques, including solution DEER spectroscopy, HDX-MS, and SAXS, likewise support the idea that Env can spontaneously transition toward more open conformations[18,20,21]. Based on these observations, we propose a model for the conformational trajectory of HIV-1 Env that encompasses a series of trimeric states: closed (A), base-relaxed closed (A$_{BR}$), moderately open (B), partially open (C), asymmetrically open (BCD, CCD, CDD), and fully open (D), defined by the state of each Env protomer (Fig. 7). In this model, Env naturally samples states A, B, C, and followed by D, in a stepwise manner. The equilibrium among these conformations varies by viral isolate: tier 2 and 3 isolates are strongly biased towards the closed state, whereas tier 1 isolates populate open

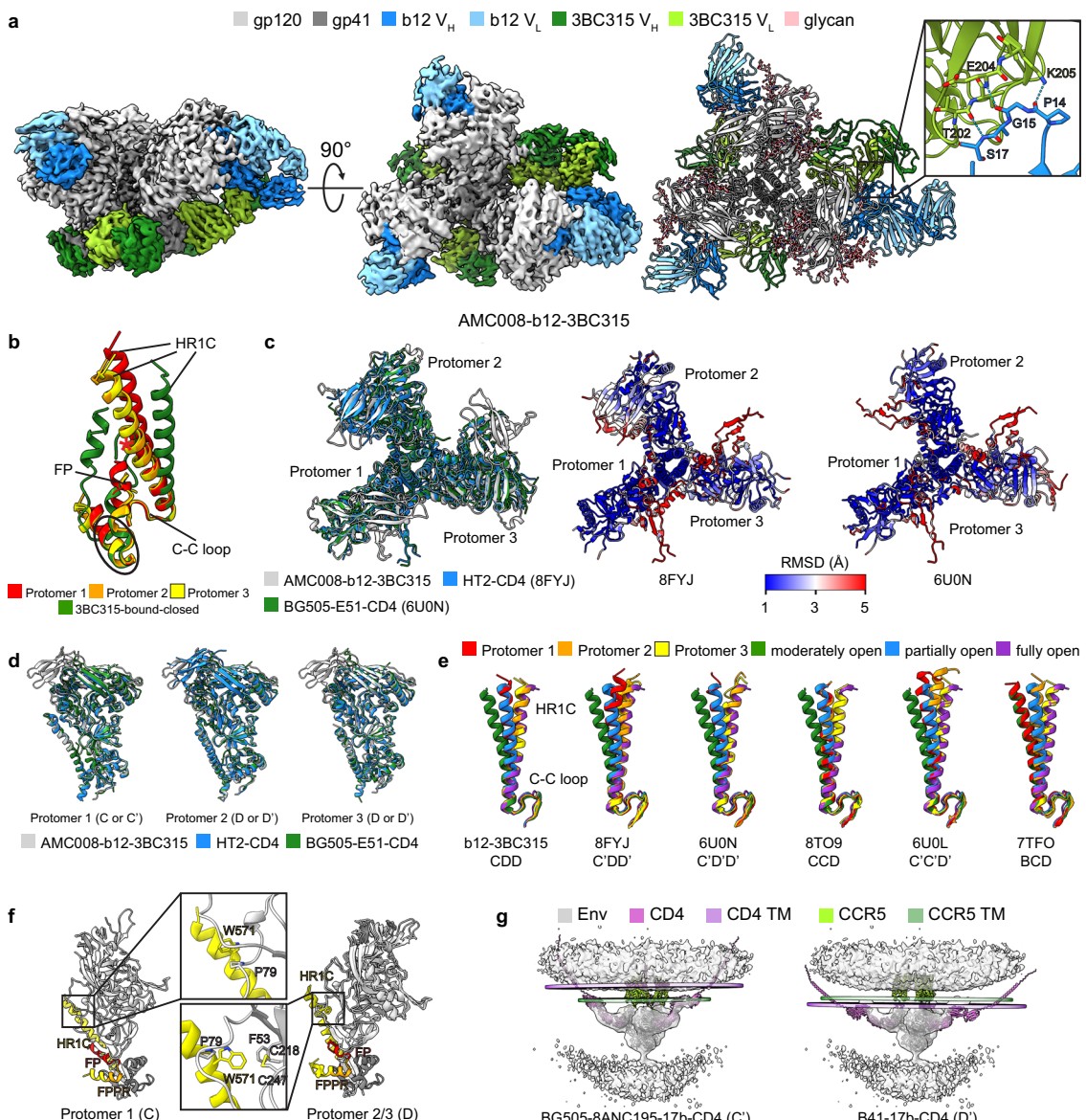

**Fig. 5 | Remodeling of AMC008 SOSIP.v4.2 upon b12 and 3BC315 binding. a** 3.7 Å cryo-EM density map (left and middle, top and side views) and atomic model (right) of AMC008 in complex with b12 and 3BC315 Fabs. A close-up view shows interactions between the 3BC315 light chain and the b12 heavy chain. **b** Comparison of the gp41s in AMC008-b12-3BC315 to the 3BC315-bound gp41 in AMC008-PGT121-VRC01-3BC315. Clashes are indicated by red * marks. 3BC315 binding epitope is circled. **c** Left: superimposition of AMC008-b12-3BC315, HT2-CD4 (8FYJ) and BG505-E51-CD4 class II (6U0N). Middle: Cα RMSD of HT2-CD4 in comparison with AMC008-b12-3BC315. Right: Cα RMSD of BG505-E51-CD4 class II in comparison with AMC008-b12-3BC315. **d** Superimposition of protomer 1 (left), 2 (middle), and 3 (right) of b12 and 3BC315-bound AMC008, two CD4-bound HT2, and three CD4-bound BG505. **e** Comparison of the protomeric HR1C conformations in asymmetric

open Envs to moderately open (AMC008-b12), partially open (6CM3), and fully open conformations (5VN3). **f** Comparison of the distinct conformations between protomeric states C and D. Close-up views show the different location and orientation of the W571 side chain. **g** Estimation of transmembrane planes of CCR5s and CD4s in states C' and D'. Models of states C' (6CM3) and D' (5VN3) Envs are docked into the cryo-ET density map (EMD-29294 [https://www.ebi.ac.uk/pdbe/entry/emdb/EMD-29294]). Models of gp120-CD4-CCR5 (6MEO) and predicted full-length CD4 are aligned to the docked Envs. CD4-bound states are denoted with'. Fabs and CD4s are hidden for better visualization in (**c, d,** and **f**). V_H, heavy chain variable domain; V_L, light chain variable domain; FP, fusion peptide; FPPR, fusion peptide proximal region; C-C loop, disulfide loop; HR1C, C-terminal region of heptad repeat 1; TM, transmembrane domain.

states more frequently. The effects of CD4 engagement further illustrate this isolate dependence. CD4 binding proceeds in a stepwise manner, with the response to the first CD4 molecule differing markedly across isolates. For BG505, a tier 2 Env, binding of a single CD4 yields a mixture of conformations, with approximately two-thirds of trimers remaining closed and one-third becoming asymmetrically open. In contrast, BaL, a tier 1b Env, becomes fully asymmetrically open upon binding just one CD4[13,14]. Furthermore, structural studies of antibody-bound states reinforce this theme. For example, T/F100 Env bound to 8ANC195 adopts a base-relaxed state $A_{BR}$, whereas the same

antibody binds BG505 in a closed state A[30]. Our AMC008 bound to 3BC315 adopts an $A_{BR}$ state, while BG505.MD39.3 bound to 3BC315 remains closed[44]. b12-bound B41 Env exhibits a fully open state D while b12-bound AMC008 exhibits the moderately open state B[10]. Altogether, these structural and biophysical data highlight that the Env conformational landscape is both isolate- and mutation-dependent, making it truly multidimensional.

Our AMC008-b12 complex is the symmetric state B Env, and it is a crucial intermediate to delineate the transition from state A to state C. For Env to transition from closed to open states, it must assume state B

to avoid the steric clash occurring at the C and D strands of the V2 loop (Supplementary Movies 2 and 3). In line with our observations, SAXS studies suggest the existence of an early intermediate state of HIV-1 Env that involves increased flexibility around the apex loop contacts[20]. And smFRET studies further show that binding of one CD4 leads to the adjacent Env protomer sampling an intermediate conformational state between the closed state and CD4-bound open state[43]. This conformational state in the protomer adjacent to the CD4-bound protomer most likely corresponds to state B of the present work. Our models of AMC008-b12 (B) and AMC008-CD4 (B') provide the structural evidence for this intermediate state.

CD4 can engage Env at each state; however, the coreceptor binding site is only accessible in CD4-bound states C' and D', but not in CD4-bound A' and B'. By CD4 and CCR5 membrane surface measurements, we tentatively conclude that the coreceptor engages Env in state D'; and states A', B', and C' are early intermediates of CD4 engagement (Fig. 5g). Thus, we hypothesize that at least one protomer of Env in state D' is required for Env transition to postfusion. A recent study reported three CD4-bound Env structures stabilized by the FP-binding antibody VRC34.01[37]. In these structures, VRC34.01 binding reorients the FP outward, decoupling its conformation from the states in our proposed model. If classified without considering FP conformation, these structures correspond to the B'B'C', B'C'C', and C'C'C' states (Supplementary Fig. 4q–s and Supplementary Table 1). These states effectively bridge the gap between the moderately open (B/B') and partially open (C/C') states in our present model. Considering the sequence homology and structural similarities of HIV-1 Env, it is likely that Envs of SIVcpz, SIVgor, HIV-2, and SIVsmm follow similar conformational trajectories[45–47]. It is important to note that this model is derived from studies of soluble, stabilized Env constructs. On the native membrane, the energy landscape and transition kinetics may differ due to the absence of the transmembrane domain and cytoplasmic tail, and the presence of artificial stabilizations. Nevertheless, our present model provides a structural framework for understanding the conformational plasticity of HIV-1 Env and serves as a valuable template for therapeutic development and immunogen design.

## Methods
### DNA constructs
Constructs for transient expression of antibodies 35O22, VRC01, b12, and PGT121 in mammalian cells were obtained from the Kulp group at the Wistar Institute (Philadelphia, PA). Construct for transient expression of soluble CD4 in mammalian cells was obtained from the Andrabi group at the University of Pennsylvania (Philadelphia, PA). A construct for the expression of antibody 3BC315 in mammalian cells was designed in-house and synthesized by a commercial vendor (Genscript). The full-length BG505 plasmid was used to make point mutations at T332N for pseudotype virus production. Plasmids expressing HIV backbone ΔEnv (pSG3), SF162, BAL, TRO11, and murine leukemia virus (MLV) control envelope were obtained from the NIH AIDS Reagents Program.

### AMC008 SOSIP.v4.2 and CD4 expression and purification
AMC008 SOSIP.v4.2 Env and furin were transiently co-transfected in Expi293F cells (Gibco; A14527) at a cell density of $2.5 \times 10^6$ cells/mL. The supernatant of the cell culture was harvested five days after

**Table 2 | Mass shift of AMC008 SOSIP.v4.2 upon binding of excess b12, 3BC315, b12 & 3BC315, VRC01, and VRC01 & 3BC315**

| Binders | MW (kDa) | ΔMW (kDa) | Fab/55 kDa |
|---|---|---|---|
| / | 338 ± 27 | / | / |
| b12 | 508 ± 43 | 170 | 3.09 |
| 3BC315 | 452 ± 42 | 114 | 2.07 |
| b12 & 3BC315 (P1) | 503 ± 40 | 165 | 3.00 |
| b12 & 3BC315 (P2) | 670 ± 36 | 332 | 6.04 |
| VRC01 | 502 ± 21 | 164 | 2.98 |
| VRC01 & 3BC315 | 568 ± 45 | 230 | 4.18 |

Increased molecular weights are normalized to the number of Fab binding by 55 kDa per Fab. Data are measured by mass photometry, and mass histograms are provided in Supplementary Fig. 6. *MW* molecular weight, *ΔMW* increased molecular weight, *P1 and P2* peak 1 and peak 2 of the mass histogram.

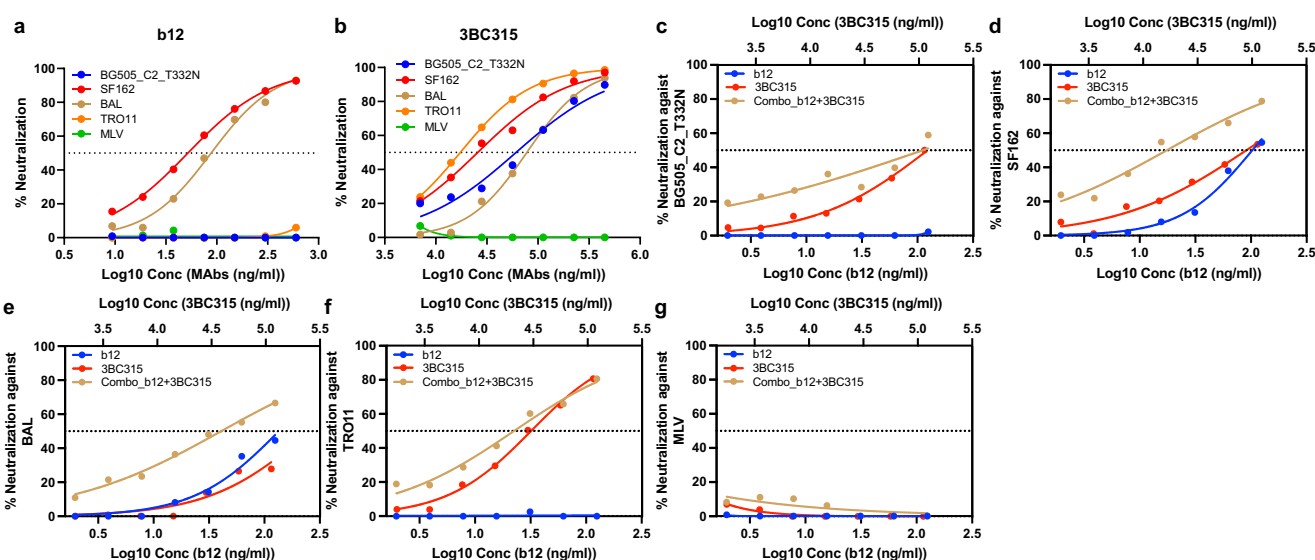

**Fig. 6 | Neutralization cooperativity of b12 and 3BC315. a-b** Neutralization of HIV pseudoviruses BG505_T332N, SF162, BAL, TRO11 or MLV by varying concentrations of (**a**) b12 and (**b**) 3BC315. **c-g** Neutralization by b12 or 3BC315 alone at predetermined concentrations and the combination of b12 and 3BC315 against HIV pseudoviruses (**c**) BG505_T332N, (**d**)SF162, (**e**) BAL, (**f**) TRO11, and (**g**) MLV. Samples were tested starting at a 1:5 dilution, followed by serial 2-fold dilutions, and neutralization curves were generated from relative luminescence unit (RLU) measurements. Percent infection was normalized to wells on the same plate with virus only and no-mAb (100%) vs cells only (0%). Each mAb-virus combination was tested in duplicate wells (n = 2 technical replicates) within a single experiment. The neutralization assay was performed four times independently; a representative experiment is shown. Source data are provided as a Source Data file.

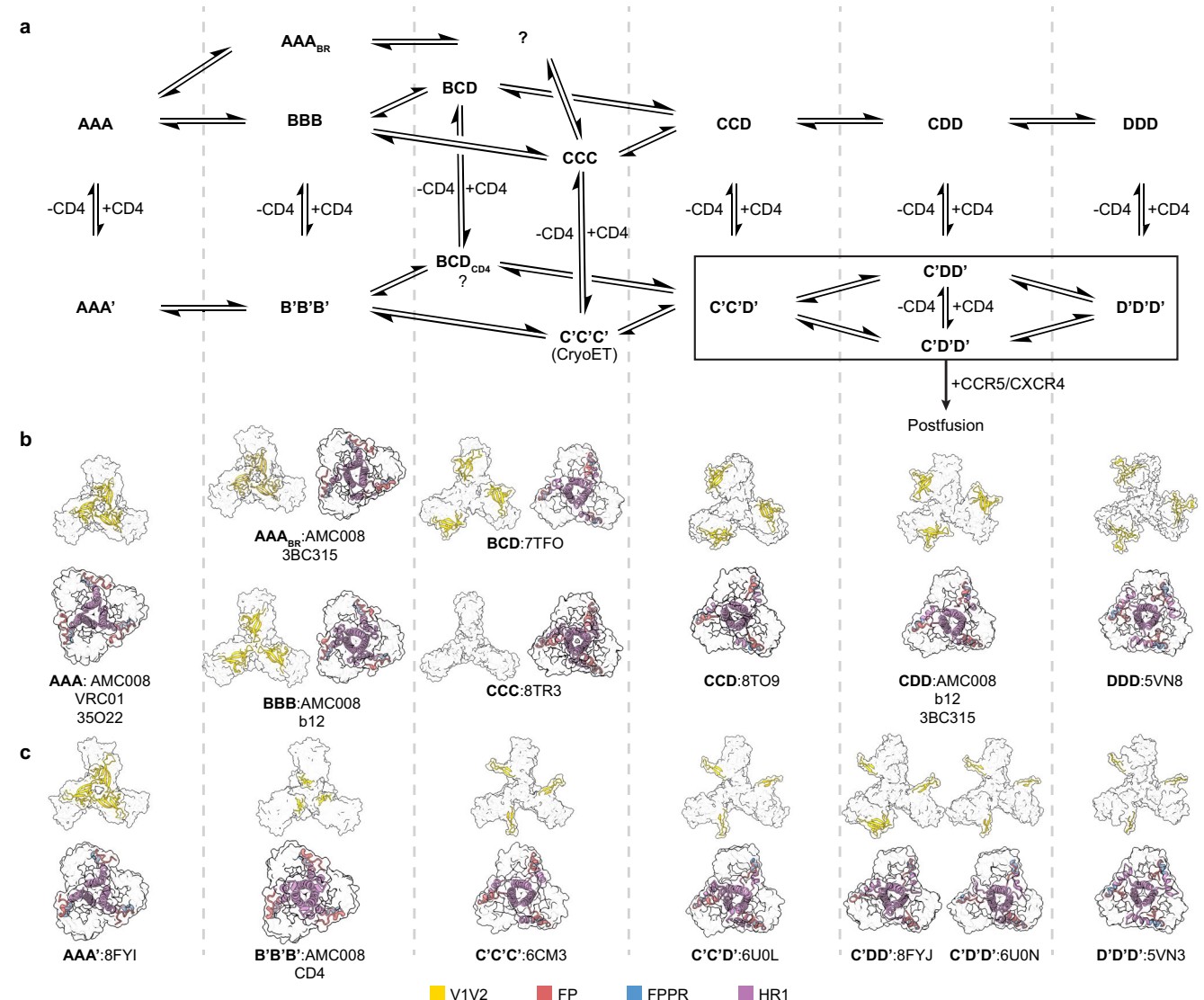

**Fig. 7 | Proposed mechanism of conformational changes from closed to fully open pre-CD4-bound and post-CD4-bound Env trimers. a** Mechanism of conformational changes from closed (AAA) to moderately open (BBB) to partially and asymmetric open (BCD, CCC, CCD, and CDD) and fully open (DDD). These conformations are capable of engaging CD4, and CD4-bound states are denoted with'. AAA$_{BR}$ denotes the closed Env states with one, two, or three base-relaxed protomers. BCD$_{CD4}$ denotes one, two, or three CD4-bound BBB or BCD states. **b** Representative models with the designated states for pre-CD4-bound conformations. **c** Representative models with the designated states for CD4-bound conformations. There are no existing models for CD4-bound BCD state. FP fusion peptide, FPPR fusion peptide proximal region, HR1 heptad repeat 1.

transfection, or once cell viability was below 50%, by centrifugation at 6000 g for 30 minutes, followed by 0.2 μm filtration and the addition of 1 mM PMSF. 0.5 mL lectin resin (Vector Laboratories) was added to the supernatant and incubated overnight. The overnight incubated supernatant was loaded and flowed through a gravity column (Bio-Rad), washed with 10 column volumes (CV) of PBS, and eluted with 5 CV of elution buffer (1 M Methyl α-Mannopyranoside in PBS). The eluent was desalted by a PD-10 desalting column (Bio-Rad) and concentrated to 1 mL by centrifugal filter with a 100 kDa molecular weight cutoff (Millipore). The lectin-purified and desalted AMC008 Env protein was a mixture of AMC008 aggregate, trimer, and monomer. The mixture was purified by SEC using a 10/300 Superose 6 Increase column (Cytiva). Fractions corresponding to the AMC008 Env trimer were collected, concentrated using a centrifugal filter with a 100 kDa molecular weight cutoff to 1 g/L, snap-frozen, and stored at -80 °C for future usage.

Soluble CD4 was transiently expressed in Expi293F cells at a cell density of 2.5×10$^6$ cells/mL. The supernatant of the cell culture was harvested five days after transfection, or once cell viability was below 50%, by centrifugation at 6000 × $g$ for 30 minutes, followed by 0.2 μm filtration and the addition of 1 mM PMSF. 0.5 mL Ni-IMAC resin (Thermo Fisher Scientific) was added to the supernatant and incubated overnight. The overnight incubated supernatant was loaded and flowed through a gravity column, washed with 10 CV of PBS, and eluted with 5 CV of elution buffer (500 mM imidazole in PBS). The eluent was concentrated to 1 mL by centrifugal filter with a 10 kDa molecular weight cutoff (Millipore) and purified by SEC using a 10/300 Superdex 200 Increase column (Cytiva). Fractions corresponding to the soluble CD4 were collected, concentrated using a centrifugal filter with a 10 kDa molecular weight cutoff to 5 g/L, snap-frozen, and stored at −80 °C for future usage.

## Antibody digestion and Fab purification

The heavy chain and light chain of antibodies were transiently co-expressed in Expi293F cells with a ratio of 3:2. The supernatants of the cell cultures were harvested seven days after transfection or once cell viability was below 50% similarly to AMC008 Env. 1 mL Protein A resin (Cytiva) was added to the supernatant and incubated overnight. The overnight incubated supernatant was loaded onto a gravity column, washed with 10 CV of PBS, and eluted with 5 CV of elution buffer (20 mM sodium citrate, 150 mM sodium chloride, pH 2.5). Eluent was collected into a 50 mL centrifugal tube containing 5 mL of 1 M pH 8 Tris to neutralize the elution buffer. Eluted antibodies were buffer-exchanged and concentrated to 5–10 mg/mL by centrifugal filtration with a 100 kDa molecular weight cutoff and stored at 4 °C.

The purified antibodies were diluted to around 1 g/L in digestion buffer (10 mM L-cysteine, 100 mM sodium acetate, 0.3 mM EDTA, pH 5.6). Papain (Millipore) was pre-incubated in digestion buffer at 37 °C for 10 min. The activated papain was added to the antibody and incubated at 37 °C overnight. Digestions were terminated by adding 3 mM iodoacetamide. The digested mixtures containing Fab, Fc, and undigested antibody were purified by Protein A resin. 1 mL of Protein A resin was added to the mixtures and loaded onto a gravity column and incubated for 10 minutes. Flow-through was collected after incubation, concentrated, and buffer exchanged to PBS in a centrifugal filter with a 10 kDa molecular weight cutoff (Millipore). 10 CV of PBS was added to wash the column. 5 CV of Protein A elution buffer was added to elute Fc and undigested antibody. Fabs were stored at 4 °C for future usage.

## Complexation and cryo-EM sample preparation

Complexes were formed by incubating AMC008 Env with Fabs or CD4 in a 1:9 molar ratio in PBS for at least one hour at 4 °C. The mixture was purified by SEC using a 10/300 Superose 6 Increase column. The fractions corresponding to the complex were collected, combined, concentrated to 1 g/L, snap-frozen, and stored at -80 °C for future use.

Complexes were thawed, diluted to around 0.1 g/L, and placed on ice for cryo-grid preparation. GF-1.2/1.3-3Au-45nm cryo-EM grids (Protochips) were glow-discharged for 30 seconds, and the glow-discharged grids were coated with graphene oxide (GO). Complexes were vitrified by applying 4 µL diluted sample to the GO-coated grid using the vitrobot Mark IV (FEI) at 4 °C and 100% humidity, followed by plunging into liquid ethane.

## Cryo-electron microscopy data collection and processing

Cryo-EM data were collected with a 300 kV Titan Krios G3i microscope equipped with a K3 Summit Direct Electron Detector camera at 81,000x magnification, a nominal dose of 58 e-/Å2, and a defocus range from −0.5 µm to −2.5 µm. Movies were processed by RELION v3.1[48] using a standard cryo-EM data processing workflow. Workflow included motion correction, CTF estimation, LoG picking, particle extraction, 2D classification, manual selection of good 2D class averages, 3D classification and refinement, and Bayesian polishing. The polished particles were imported into cryoSPARC[49] for Non-uniform Refinement. Consensus maps of AMC008-PGT121-VRC01-3BC315, AMC008-VRC01-35O22, AMC008-b12, AMC008-CD4, AMC008-3BC315, and AMC008-b12-3BC315 were generated with the workflow mentioned above (Supplementary Figs. 1, 3, and 5).

In the AMC008-PGT121-VRC01-3BC315 dataset, we observed zero or one 3BC315 binding to the AMC008 SOSIP.v4.2 Env. We performed a classification focused on the 3BC315 binding site with three classes. Two classes that had 3BC315 binding were combined and reconstructed to the final AMC008-PGT121-VRC01-3BC315 map (Supplementary Fig. 1b).

In the AMC008-CD4 dataset, we observed one to three CD4s binding to AMC008 SOSIP.v4.2. We simulated three density maps of one, two, and three CD4s binding to AMC008 SOSIP.v.4.2 and performed Heterogeneous Refinement with the simulated maps. The

resulting 3D classes of AMC008-CD4 were all in the same conformational state despite the binding occupancy of CD4. We reconstructed the final map with particles of the three CD4-bound class and imposed C3 symmetry, as the three CD4-bound class was most populated and had the best view distribution. To investigate the V1V2V3 conformation of AMC008-CD4, we transferred particles of the three CD4-bound class back to RELION and performed a 3D classification focused on the Env apex with four classes. Two of the classes with density definition on Env apex were combined and imposed C3 symmetry to reconstruct a map with better density on Env apex (Supplementary Fig. 5a).

In the AMC008-VRC01-35O22 dataset, we observed exclusively three VRC01 Fabs binding to the Env, while one to three 35O22 Fabs bound to the AMC008 SOSIP.v4.2 Env when processing the cryo-EM data. Reconstructing each complex population with different 35O22 Fab binding occupancy, we found that the binding of 35O22 did not alter the AMC008 SOSIP.v4.2 Env confirmation and only subtle differences were observed around the 35O22 epitope. After applying multiple data processing schemes, we found that combining populations with two or three 35O22 Fab binding occupancy and imposing C3 symmetry gave the best 3D reconstruction, considering both the global and local resolutions at the Env base (Supplementary Fig. 1c).

In the AMC008-3BC315 dataset, we observed one or two 3BC315 binding to the AMC008 SOSIP.v4.2, and the ratio of the two populations was about 1:1. Reconstruction of the two classes showed dramatic conformational changes around the 3BC315 epitope. We performed a classification focused on the 3BC315 binding site with ten classes. We only included particles with high confidence in 3BC315 binding occupancy for the final round of reconstruction (Supplementary Fig. 3a).

## Model building and refinement

Crystal structures of CD4 (PDB: 1WIO), VRC01 (PDB: 4LST), 35O22 (PDB: 4TOY), PGT121 (PDB: 4JY4), b12 (PDB: 1HZH), and 3BC315(PDB: 5CCK) were used as reference models. A reference model of AMC008 Env was generated by homology modeling using Modeler[50]. Reference models were rigid-body docked into the density maps in UCSF Chimera[51] and manually rebuilt in Coot[52]. N-linked glycans were added to the models according to the N-linked glycosylation consensus sequence and the density maps. The manually rebuilt models were further refined using Rosetta FastRelax[53]. Refined models were reinspected and adjusted in Coot and refined by Rosetta again. These steps were repeated until no significant improvement was observed. Model geometry was validated by MolProbity[54], glycan conformation was validated by Privateer[55], and model fit-to-map was validated by EMRinger[56].

## Mass photometry

Complexes for mass photometry were prepared by mixing AMC008 Env with Fabs in a 1:9 molar ratio in PBS and incubating on ice for 30 minutes. Samples were diluted in PBS to optimal concentrations right before data collection. Data were collected by a TwoMP mass photometer (Refeyn) in default settings for 60 seconds and were calibrated by a ladder comprising bovine serum albumin (66 kDa), β-amylase (224 kDa), and thyroglobulin (670 kDa). Raw data were analyzed using DiscoverMP software to generate mass histograms.

## Cell lines and transfections for neutralization assay

HEK 293 T cells (ATCC; CRL-3216) and TZM-bl cells (NIH AIDS Reagents Program) were maintained in DMEM (ThermoFisher) supplemented with 10% heat-inactivated fetal bovine serum (Atlas Biologicals). Pseudotype viruses were produced as previously described[57]. Briefly, HEK 293 T cells were co-transfected with 4 µg of a plasmid encoding the desired Env protein and 8 µg of a plasmid expressing the HIV-1 backbone Δ Env (pSG3ΔEnv − NIH AIDS Reagents) using GeneJammer (Aglient). Forty-eight hours after transfection, cell supernatants were

harvested, filtered using a 45 μm filter, aliquoted, stored at −80 °C and titered.

### Neutralization assay

Pseudotyped viruses were titered to yield 150,000 RLU after 48 h of infection with TZM-BL cells[57]. Monoclonal antibodies (MAbs) were serially diluted either individually or mixed together as combination at predetermined concentrations in 96-well plates followed by incubation with respective pseudotyped viruses before adding 10,000 TZM-BL cells (NIH AIDS Reagent Program) per well with dextran (Thermo-Fisher). Forty-eight hours post incubation, media was removed, and cells were lysed using BriteLite luciferase reagent (Revvity). Luminescence was then measured using the Synergy2 plate reader (BioTek Instruments).

### Statistical analysis

All statistics and calculations were performed using GraphPad Prism 10.0. MAb titer was determined for 50% virus neutralization (ID50), values were computed and graphed with a nonlinear regression model of percentage neutralization vs log10 concentration of MAbs.

### Reporting summary

Further information on research design is available in the Nature Portfolio Reporting Summary linked to this article.

## Data availability

The atomic models were deposited to the Protein Data Bank (PDB) under accession codes 9NBT (AMC008-VRC01-35O22), 9NBY (AMC008-PGT121-VRC01-3BC315), 9YQO (AMC008-PGT121-VRC01), 9NC0 (AMC008-b12), 9OAJ (AMC008-CD4), 9NC3 (AMC008-b12-3BC315), 9NC6 (AMC008-3BC315 (2x)), and 9NC8 (AMC008-3BC315 (1x)). The cryo-EM maps were deposited to the Electron Microscopy Data Bank (EMDB) under accession codes EMD-49236 (AMC008-VRC01-35O22), EMD-49238 (AMC008-PGT121-VRC01-3BC315), EMD-73342 (AMC008-PGT121-VRC01), EMD-49239 (AMC008-b12), EMD-70287 (AMC008-CD4), EMD-49240 (AMC008-b12-3BC315), EMD-49241 (AMC008-3BC315 (2x)), and EMD-49242 (AMC008-3BC315 (1x)). PDB entries 1HZH, 1WIO, 4JY4, 4LST, 4TOY, 5CCK, 5I8H, 5VN3, 5VN8, 6CM3, 6MEO, 6NQD, 6U0L, 6U0N, 6VRW, 7LO6, 7LOK, 7SQ1, 7TFN, 7TFO, 8FYJ, 8TO9, 8TR3, 9D8Y, 9D90, and 9D98, used in this study are available in the PDB. EMDB entry EMD-29294, used in this study is available in the EMDB. Source data are provided as a Source Data file. Source data are provided with this paper.

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

## Acknowledgements

We thank Thomas Klose at the Purdue University Cryo-EM facility and Ruben Diaz Avalos at the La Jolla Institute for their assistance in sample screening and data collection. We thank Boyu Yin for his assistance in mass photometry experiments. This work was funded by W.W. Smith Charitable Trust grant A2404 (to J.P.), NIH grant U19 AI166916 (to D.B.W. and J.P.). Funding sources were not involved in the design of this study, the collection and analysis of data, the decision to submit, or the preparation of the manuscript.

## Author contributions

J.C. and J.P. conceived experiments. J.C. produced protein samples. J.D. collected cryo-EM data. J.C., J.D., Z.J.L., and J.P. processed the cryo-EM data. J.C. built atomic models. J.C. and Z.J.L. performed mass photometry experiments. S.G. and R.S. performed pseudovirus neutralization assays. J.C. analyzed and interpreted the data. J.C., Z.J.L., and J.P. wrote the manuscript draft. S.G. contributed to the manuscript draft. All authors reviewed and commented on the manuscript. J.P. and D.B.W. supervised the work and acquired funding.

## Competing interests

D.B.W. notes several possible competing interests, which are managed by the Wistar General Council Office COI committee. These include consulting, BOD service, speaking, which can include in-stock or monetary remuneration, and specific SRAs. Inovio Pharmaceuticals (BOD, consultant and SRA); AstraZeneca (speaker/consultant); Geneos (consultant & SRA); and possibly others that are managed by the Wistar COI Committee. D.B.W. is a member of the International Society for Vaccines, AAI, ASGCT, and AAAS, among other scientific societies. He also serves on NIH and NCI study sections and similar activities for other agencies. The remaining authors declare no competing interests.
