## [Transparent Peer Review file · Nature Communications]

Conformational Landscape of HIV-1 Env from Closed to Fully Open

Corresponding Author: Professor Jesper Pallesen

Version 0:

Reviewer comments:

Reviewer #1

(Remarks to the Author)

This study reports cryo-EM structures of HIV-1 AMC008 envelope (Env) in complex with various combinations of 3BC315, b12, VRC01, 35O22, and CD4, identifying several conformational states including a previously uncharacterized, moderately open, intermediate (between closed and partially open forms). The authors propose a classification system (states A-D) based on gp120 rotation angles and gp41 rearrangements. The work demonstrates that antibodies 3BC315 and b12 exhibit enhanced binding when used together, attributed to antibody-antibody interactions and conformational changes that increase 3BC315 occupancy from 1-2 to 3 copies per trimer. Mass photometry and neutralization assays support a form of synergistic binding, showing enhanced neutralization effects. The structures reveal that conformational transitions require passage through intermediate states to avoid steric clashes, particularly involving V2 loop contacts. The accompanying movies are helpful in illustrating this. While the study provides new structural intermediates and a classification approach for Env conformations, several conclusions rely strongly on modeling and assumptions that antibody bound conformations represent natural transition conformations. The conclusions are of interest to the general HIV field, though the steps of the conformational changes are somewhat incremental and complicated, the results are important. The findings contribute to understanding Env conformational dynamics but would benefit from addressing the issues below. I recommend acceptance with minor revisions.

Major

The movies/conformational transitions assume static V2 domain structures. The intermediate this is based on from the b12-bound structures is likely an artifact of b12 stabilizing the V2 domain while the rest of the gp120/gp41 adopts a partially opened form. The much more likely scenario is that upon binding CD4 the V2 domain opens either up or extended towards the CD4 and then the twists/transitions in gp120/gp140 regions observed here occur, the movies are a little misleading in that respect. This scenario needs to be investigated/outlined in detail along with the one currently proposed.

Data or discussion supporting this as an actual pathway as opposed to antibody induced conformations would be helpful.

The data supporting the cooperative binding and neutralization, from a conformational preference standpoint induced by b12, is solid. But while the Fabs make contact through the framework region of b12 and FC region of 3BC315, claims that this contact is more than incidental is not really supported. 2 of the 3 Fabs don't show interactions and no Fab:Fab binding studies were done. Was there any SHM on b12 to support this? If not, wouldn't it interact with nearly any other Fab? If there was SHM then it might suggest b12 co-evolved to bind a similar antibody which would be of more general interest. As it's presented now, it just looks like they happen to touch because they're close to each other. This is similar to how H-bonds can be observed with some glycans and HIV-1 antibodies even though the same antibodies bind better when the glycan is absent. One potential way to support the Fab:Fab binding would be to run 3D flexibility on the density and if it was observed the two antibodies stayed together while all other regions moved around.

Minor

Based on the proposed conformational changes here, it would be helpful to see the three C3 structures built and refined in C1 to note any small differences in asymmetry, particularly for discordance in structures in the 540-560 range, the FP, and for glycan heterogeneity.

More analysis/discussion on potential glycan effects (or no effect) on the lower stoichiometry of 3BC315 would be helpful.

Somewhat similar to what is shown for b12.

Figure 6 is just difficult to interpret. I appreciate that the authors are trying to depict a very complicated system but any effort to make this a little cleaner or more interpretable would be appreciated.

While speculative, I'd like to see discussion of any other commonalities in other viral fusion proteins that this might potentially apply to. This would help with the general impact by extending the conformational pathway to other viruses.

Line 540 "PCT121" should be PGT, "Rn32" should be RnS2

Reviewer #2

(Remarks to the Author)

Jiayan Cui and colleagues study the opening of the HIV-1 Env trimer by CD4 receptor for fusion. Because the HIV-1 Env is a master of immune evasion, the mechanism of how Env is opened is of great importance, not just for the understanding of these viral fusion machines, but also for immunogen design to generate vaccines that can neutralize this virus. To gain structural insights into the mechanism of Env opening, the authors engineered a stabilized SOSIP.v4.2 trimer based on the HIV-1 strain, AMC008, which appears somewhat resistant to opening. Using this Env and a panel of neutralizing antibodies along with soluble CD4 receptor, the authors determined several Env structural intermediates and asymmetric states during the transition from closed to open Env.

For instance, in complex with the VRC01 and 35O22 Fabs, the authors determine the structure of a closed Env (designated AAA). Using the Fab 3BC315 (binding gp41-gp41 interface), the authors next identify a base-relaxed AAABR Env configuration. Using the CD4 binding site targeting antibody b12, the authors identify a partially open (more closed than B41-b12, hence designated "moderately open" Env, BBB), and with CD4 a moderately open configuration (B'B'B'). These structures are compared to partially open CCC and fully open DDD, which is the most open Env configuration that likely moves closest towards the membrane to engage the coreceptor. In summary, the study identifies novel conformational states of Env, from the fully closed, a "base-relaxed" state, a "moderately open" state, to partially open, then finally fully open, as well as asymmetric configurations thus revealing a complex conformational landscape for the dynamic opening of HIV-1 Env. The report is incredibly comprehensive and I have only minor suggestions that the authors could consider.

Looking at the final model for the HIV-1 Env opening presented in the Fig. 6a scheme, would it be fair to mention in the discussion that this is consistent with a multidimensional conformational landscape with parallel paths where each HIV-1 isolate, and each Env mutant might follow a slightly different path? It has been speculated that Env features a multi-dimensional landscape, and I have never seen this documented in a more impressive way than in Fig. 6a.

The authors present several Env structures bound to b12, representing different opening states from moderately open to partially open and fully open. As expected, gp120 models in these structures appear almost identical since b12 Fabs bind to the same epitope across all states. Furthermore, the HR1C helices of gp41 align well across these moderately, partially, and fully open structures (Fig. 2i), suggesting that changes in the gp41 bases and gp41-gp120 interfaces play a greater role in the overall opening of Env. This is likely important for the CD4 engagement on membrane-bound Env and may involve tilting of the trimer. However, the authors places heavy emphasis on HR1C direction (Fig. 2e-h and Fig. 4e), which could mislead readers into thinking HR1C movement drives Env opening. Maybe the authors could re-analyze the differences in gp41 base by superimposing the structures on HR1C and focusing the discussion on base rearrangements? It may also help to explain the variable stoichiometry of 3BC315 binding observed across different complexes.

The authors determined the structure of AMC008.SOSIP.664-b12-3BC315 complex and observed an interaction between b12 and 3BC315 Fabs, which facilitates increased accessibility of 3BC315. This was further supported by mass photometry, which revealed increased numbers of bound 3BC315. A synergistic neutralization effect was observed on b12-sensitive strains. However, such an effect was minimal on b12-resistant strains, where the IC50 of 3BC315 remained similar in the presence or absence of b12. These observations suggest that b12-bound state favors 3BC315 binding, but the reverse may not be true. However, the authors interpret this interaction as cooperative binding. The authors could perform flow cytometry with sequential antibody addition (e.g., adding b12 followed by 3BC315, and vice versa) to assess whether each antibody enhances the binding efficiency of the other. Yet, again, this is not central to the main message of this manuscript.

Although the structural interpretations are well supported, some figures could be clearer. For instance, in Fig. 1d, a more zoom-out view could help to show the overall context of the formed cavity; Fig. 1f, presenting the structures in separate groups may better to highlight the differences; Fig. 2i, indicating the superposition on HR1C would improve clarity; Figs. 3c, 4b, 4e, additional labels or annotations would make these figures more accessible to a broader audiences, etc. For the key model Fig. 6, would it be helpful to have a side view row to illustrate the mechanism of opening?

The literature review is comprehensive, and then some readers may wonder why some papers were not discussed. For instance, could a recent paper by Bhisem Thakur et al., and Priyamvada Acharya (PMID: 40382314) be discussed? Also, Youdong Mao who's reputation had been damaged from his early structural work on HIV-1 Env, has however solved recent structures that follow a mainstream structural approach as well as mainstream thinking of Env opening, includes insights into asymmetric states and the MPER peptide (PMIDs: 37202420, 40089599).

1. Line 234: for consistency, use "AMC008-b12-3BC315" throughout.
2. Line 303: it should be "Fig. 5e,f"

Reviewer #3

(Remarks to the Author)

The manuscript titled "Conformational Landscape of HIV-1 Env from Closed to Fully Open" by Cui et al., determines multiple new structures of the HIV-1 Env defining conformational states in the context of gp120/gp41 targeting antibody 3BC315. The paper provides several new structures of different HIV-1 Env conformations, both in the closed and partially open states. Additionally, they provide structure-function insights into the synergy between antibodies b12 and 3BC315. The structural determinations are well done providing atomic level details of interaction interfaces and local conformations. The mass photometry and neutralization experiments showing synergy with b12 and 3BC315 are nicely paired with the structural analysis. HIV-1 Env conformations have been extensively studied over many years. This study adds valuable information to the vast knowledge that has already accumulated on this topic.

The authors should address the following:

1. Newer literature on the topic, especially studies that are highly relevant to the results and conclusions on this paper, should be better integrated in the analysis. Recently, a study was published (PMID: 40382314) that reported on the "Conformational trajectory of the HIV-1 fusion peptide during CD4-induced envelope opening". The authors have included three structures from the study in Extended Data Table 1 but should also integrate within the analysis in the main text (and cite the published study). Some places where the suggested integration can happen:
 - a. Line 133-134: The authors state that: "Taken together, these results underscore FP conformational plasticity across closed, partially open, and fully open Env states." The conclusions in the published study mentioned above are highly synergistic with this stated conclusion, reaching essentially the same conclusion with a different gp120/gp41 interface binding antibody (the FP directed VRC34.01). This would be a great place to refer to the published study and align the conclusions.
 - b. The structures (at least 9D90, 9D98 and 9D8Y) should be included in Extended Data 3 (and described in the text along with the comparisons with the other published structures).
 - c. In the section titled "CD4-bound moderately open AMC008" (line 177) authors should include comparison with CD4-bound structures: 9D8Y, 9D98 (PMID: 40382314). Also, include these structures in the comparisons described in lines 234-238.
 - d. Figure 3f: the authors indicate State C' as having a buried FP helix based on the 6CM3 structure, but this description does not consider new data that shows that this state geometry can accommodate the FP in both a buried configuration (as in the 8ANC195-bound 6CM3 structure) and in an accessible/exposed configuration (as in the VRC34.01-bound 9D90 structure). By not incorporating the newer data into their analysis, nuances that have enriched our knowledge of these states are being lost. Authors should revise.
5. line 49-50: "CD4 engagement begins with the binding of one CD4 molecule, resulting in an asymmetric open or closed conformational state of Env." This sentence is confusing and should be rephrased. I think what the authors mean here is that CD4 can bind to both closed and open Env conformations. The "result" of CD4 binding, however, is Env opening.
6. Lines 53-54: "Notably, cryo-ET reveals that three CD4-bound Env adopts a partially open state distinct from the fully open state typically observed by cryo-EM11,14". This sentence is only partially correct. While the cited paper (PMID: 37993716), captures a state that is partially open, earlier cryo-ET studies (PMID: 18668044), albeit at lower resolutions, described open structures that aligned well with the open structure of B41 Env bound to CD4 and 17b resolved by single particle cryo-EM analysis (see Extended Figure 3a in PMID: 28700571). Differences in Env used or the specific membrane context used in the studies could be responsible for the differences. Authors should revise the sentence to be better reflective of the published data in the literature.
7. In line 64 and 67, 3BC315 is described as "targeting the gp41-gp41 interface"
In line 96-97, its described as "3BC315 binds predominantly to one gp41 subunit but exhibits additional interactions with gp120 (same protomer) and adjacent gp41."
Line 106: "3BC315 CDRH3 loop inserts into a cavity at the gp120-gp41-gp41 interface"
Given that 3BC315 contacts both gp120 and gp41 subunits, it may be better to call it a gp120-gp41 interface targeting antibody. In any case, authors should examine these diverse descriptions in the manuscript and try to homogenize.
8. Figure callouts are out of sequence. For example, Extended Data Fig. 5d is called out first, then Extended Data Fig. 2d, then figure 1c and so on. This makes it difficult for the reader having to go back and forth through the figures. The figures and callouts should be re-organized to go in sequence.
9. The section titled: "3BC315-bound base-relaxed AMC008" begins with the conclusion "The AMC008-PGT121-VRC01-3BC315 structure reveals remodeling of the Env base as compared to occluded closed Envs9." It would be better instead if the results were presented first, followed by the conclusion.
10. Lines 113-115: "Superimposition of the AMC008-PGT121-VRC01-3BC315 and occluded closed AMC008-VRC01-

35O22 structures demonstrates dramatic displacements by more than 5 Å in epitope-associated structural elements upon 3BC315 binding (Fig. 1b).” Here the authors compare the 3BC315 bound structure with the 35O22-bound structure to draw conclusions about the structural rearrangements caused by 3BC315. Since, 35O22 is itself a gp120/gp41 interface targeting antibody, it would be helpful to add a comparison of 35O22 (perhaps in the Supplement) with a closed Env structure that does not have an antibody bound at the gp120/gp41 interface.

12. Lines 157-159: “Integrating gp120 openness, gp41 conformation, and CD4-binding status, we categorized Env states as follows: A (closed), ABR (base-relaxed closed), B (moderately open), C (partially open), C’ (CD4-bound partially open), D (open) and D’ (CD4-bound open).” This sentence occurs somewhat prematurely without adequate details about the rationale behind the nomenclature. Those details come later in the manuscript (lines 204-217), so the authors should consider some re-organization/clarifications.

On the same note, many of these states have been described before in the literature and were named a certain way. It would be better to either adhere to the nomenclature that these states are known by in published literature or, when warranted, to provide clearly stated rationale for renaming them. For example, in lines 126-127, authors states: “In contrast, in occluded fully open B41-b12 and CD4-bound accessible fully open B41-17b-CD4”, calling both the b12-bound B41 structure and the CD4/17b-bound B41 structure “fully open” is confusing. The b12-bound B41 Env structure was previously described as “open occluded” (PMID: 35136084).

To summarize, it is highly recommended that authors examine their system of nomenclature, align it with published literature, changing and redefining only when necessary.

13. There is opportunity for improving the analysis of the b12-bound states. In the study, the authors identified a moderately open b12-bound Env and have compared it to the the b12-bound B41 Env structure (more open). Authors should also include in their analysis the structure of b12 bound to an engineered closed Env (PMID: 38306419; Figure 5 E, F). Additionally, they should also refer to the analysis in PMID: 28700571, Extended Figure 5F. All these structures, published and the ones determined in this study, taken together, show that b12 can bind to Envs in different conformations, although a more open conformation is preferred.

14. Line 240-241: “Despite these similarities, gp120s of AMC008-b12-3BC315 lack the features of a CD4-bound accessible open conformation (Fig. 4d).” – authors should state here what these features are that differentiate the states.

15. It is unclear what Figure 4g is showing that is new and derived from the results of this study. These results and conclusions were described in previous studies where these structures were published, in particular, in PMID: 37993716. This section should be re-organized to perhaps include in the discussion instead of stating it as a result of this study. The EMDB # of the cryo-ET map should be included in the figure legend.

16. Figure 2c, legend: “b12 cannot bind AMC008 in closed conformation due to clashes of N197 and N301 glycans with b12. Clashes are indicated by red * marks. d Binding of b12 requires minimum openness of AMC008 gp120 to accommodate N197 and N301 glycans.” Authors should rephrase this since modeled clashes with glycans do not necessarily preclude binding as glycans can move to accommodate binding (as the authors have themselves shown).

17. Figures showing local resolution of the cryo-EM maps should be included in the Supplement.

Version 1:

Reviewer comments:

Reviewer #1

(Remarks to the Author)

The authors have addressed my major concerns. I recommend acceptance but provide the following comments that I hope will be taken into account to help improve the final manuscript.

I noted this in my initial summary but maybe should have included it more specifically as an enumerated issue to address. The authors make sweeping assumptions throughout the manuscript about antibody-bound, stabilized states on soluble Env representing natural intermediates of Env that are membrane-bound and lack stabilization. I’m even on their side for the most part but the readers need to be better informed of the assumptions and limitations presented here compared to membrane-bound Env on a virion. This could be with caveats in the introduction and the discussion, but it should be thorough.

Authors have trimmed down the relevance of the incidental direct interaction of the two antibodies in the results but have left it in the abstract. While it’s technically correct, authors may want to focus on the more relevant conformational synergistic

results from binding the two antibodies.

Reviewer #2

(Remarks to the Author)

The reviewers have addressed our previous concerns and we support publication!

Reviewer #3

(Remarks to the Author)

The authors have thoroughly addressed reviewer comments. I have no further questions.

We are grateful for the Reviewers' overall positive evaluation of our manuscript as well as their recommendations to accept our manuscript for publication with minor revisions. We have addressed all issues raised by the Reviewers and by doing so, we believe our manuscript has been significantly improved. Please find enclosed our point-by-point response to Reviewers' evaluation.

Reviewer #1 (Remarks to the Author):

This study reports cryo-EM structures of HIV-1 AMC008 envelope (Env) in complex with various combinations of 3BC315, b12, VRC01, 35O22, and CD4, identifying several conformational states including a previously uncharacterized, moderately open, intermediate (between closed and partially open forms). The authors propose a classification system (states A-D) based on gp120 rotation angles and gp41 rearrangements. The work demonstrates that antibodies 3BC315 and b12 exhibit enhanced binding when used together, attributed to antibody-antibody interactions and conformational changes that increase 3BC315 occupancy from 1-2 to 3 copies per trimer. Mass photometry and neutralization assays support a form of synergistic binding, showing enhanced neutralization effects. The structures reveal that conformational transitions require passage through intermediate states to avoid steric clashes, particularly involving V2 loop contacts. The accompanying movies are helpful in illustrating this. While the study provides new structural intermediates and a classification approach for Env conformations, several conclusions rely strongly on modeling and assumptions that antibody bound conformations represent natural transition conformations. The conclusions are of interest to the general HIV field, though the steps of the conformational changes are somewhat incremental and complicated, the results are important. The findings contribute to understanding Env conformational dynamics but would benefit from addressing the issues below. I recommend acceptance with minor revisions.

We thank the Reviewer for their positive evaluation of our manuscript and their constructive comments. We further thank the Reviewer for recommending publication of our work. Our point-by-point comments addressing the issues appear below.

Major

The movies/conformational transitions assume static V2 domain structures. The intermediate this is based on from the b12-bound structures is likely an artifact of b12 stabilizing the V2 domain while the rest of the gp120/gp41 adopts a partially opened form. The much more likely scenario is that upon binding CD4, the V2 domain opens either up or extended towards the CD4, and then the twists/transitions in gp120/gp140 regions observed here occur; the movies are a little misleading in that respect. This scenario needs to be investigated/outlined in detail along with the one currently proposed.

Data or discussion supporting this as an actual pathway as opposed to antibody induced conformations would be helpful.

We thank the Reviewer for presenting their suggested pathway and we analyzed our data accordingly. We emphasize that the moderately open Env states exemplified by AMC008-b12 or AMC008-CD4 (states B/B') are distinct and less open than previously reported partially open states (C/C'). The Reviewer suggests that V2 may open up; we do see more flexibility in this region, but we do not observe exposure of the underlying co-receptor binding site.

As we summarize in the manuscript, the gp41 conformations of states B (AMC008-b12) and B' (AMC008-CD4) are almost identical with the most notable gp120 difference between the two states being the movement of $\beta 20/\beta 21$ upon CD4 binding to accommodate the insertion of the CD4 F43 residue. The similarity between states B and B' indicates that state B is parallel to state B', like states C/C' or D/D'.

The purpose of the movies (transitions from AAA to CDD, or from AAA over BBB to CDD) is to demonstrate that state B is a crucial intermediate between states A and C/D during stepwise Env opening. Due to the vast similarity of states B and B', this conclusion does not change for CD4-bound state B' being an intermediate between states A' and C'/D', as binding of CD4 only induces additional conformational changes on gp120, not on gp41.

Although our study provides a first reported of symmetric moderately open conformational Env states BBB/B'B'B', protomeric states B/B' have been observed before in asymmetric open Envs (Extended Data Table 1, states BCD (PDB: 7TFN and 7TFO), B'B'C' (PDB: 9D8Y), and B'C'C' (PDB: 9D98)), though not identified as moderately open by the authors, indicating that our observed moderately open states B/B' are not artificial or isolated examples.

We agree with the reviewer that the binding of b12 would possibly stabilize the V2 domain, resulting in a more resolved apex in the AMC008-b12 density map than in the AMC008-CD4 map, as the contacting area of b12 to Env is significantly larger than that of CD4. A similar scenario is that the V2 loop and Env apex are usually more well-resolved when a coreceptor binding-site antibody is co-complexed with CD4 and Env. Thus, we cannot conclude that the binding of CD4 destabilizes the Env apex; however, we can conclude that opening of Env destabilizes the Env apex, as the interprotomer contacts at the Env apex are disrupted during Env opening, and the binding of CD4 or b12 stabilizes the Env apex.

In accordance with the Reviewer's comment, we edited the results as follows:

Lines 219-222: CD4 binding induces state-specific changes: it leaves state A' unaltered relative to state A, remodels $\beta 20/\beta 21$ strands in state B', displaces V1V2 while forming the $\alpha 0$ helix and the 4-stranded bridging sheet to expose the coreceptor binding site in states C'/D' (Fig. 4f).

The data supporting the cooperative binding and neutralization, from a conformational

preference standpoint induced by b12, is solid. But while the Fabs make contact through the framework region of b12 and FC region of 3BC315, claims that this contact is more than incidental is not really supported. 2 of the 3 Fabs don't show interactions and no Fab:Fab binding studies were done. Was there any SHM on b12 to support this? If not, wouldn't it interact with nearly any other Fab? If there was SHM then it might suggest b12 co-evolved to bind a similar antibody which would be of more general interest. As it's presented now, it just looks like they happen to touch because they're close to each other. This is similar to how H-bonds can be observed with some glycans and HIV-1 antibodies even though the same antibodies bind better when the glycan is absent. One potential way to support the Fab:Fab binding would be to run 3D flexibility on the density and if it was observed the two antibodies stayed together while all other regions moved around.

We thank the Reviewer for these comments and suggestions. To clarify, we report interaction between one copy of b12 domain V_H and one copy of 3BC315 C_L. In response to Reviewer's comments, we ran IgBLAST of the b12 heavy chain against the IGMT human germline gene database and confirmed that there is no SHM on the b12 FR1 region (KPGASV) that is contacting the 3BC315 C_L domain. Thus, we agree with the reviewer that the Fab:Fab binding is likely incidental; at least not a consequence of antibody co-evolution.

As per Reviewer's suggestion, we conducted a 3D Variability analysis on AMC008-b12-3BC315 using CryoSPARC. The resulting volume series demonstrates that the b12 and 3BC315 Fabs observed in contact in the density map stay in contact throughout the density map series, while the other two steric pairs of b12/3BC315 Fabs are moving and contacting transiently, indicating that b12/3BC315 Fab:Fab binding, is veritable (Supplementary Movie 1).

Taken together, we confirm that while the observed Fab:Fab interactions between b12 and 3BC315 are not a result of co-evolution (SHM), the interaction contributes to the positive cooperativity of b12 and 3BC315. However, we agree with the Reviewer that our observed cooperativity between b12 and 3BC315 is likely mainly driven by the shared binding preference by both b12 and 3BC315 to the more open Env states.

We edited the results section accordingly:

Lines 230-235: Heavy chain framework region (C_{H1}-domain-FR1) of one b12 Fab interacts with light chain framework region (C_L domain) of one 3BC315 Fab indicating a mechanism to enhance binding of the two antibodies by cooperativity (Fig. 5a)^{35,36}. 3D variability analysis of the AMC008-b12-3BC315 density map confirmed these interactions, revealing that one b12/3BC315 Fab pair interaction is stable, the other two engage in transient contacts, indicating a cooperative binding mechanism (Supplementary Movie 1).

and

Lines 287-289: Moreover, as described above, simultaneous binding of b12 and 3BC315 facilitates stabilization of the complex through multiple hydrogen bonds formed between their constant framework regions (Fig. 4a).

Minor

Based on the proposed conformational changes here, it would be helpful to see the three C3 structures built and refined in C1 to note any small differences in asymmetry, particularly for discordance in structures in the 540-560 range, the FP, and for glycan heterogeneity.

We thank the Reviewer for this suggestion. To address this issue, we reprocessed the density maps of AMC008-CD4, AMC008-b12, and AMC008-VRC01-35O22 in C1 and built the corresponding atomic models. Alignments of the C1 structures against C3 structures of AMC008-CD4 and AMC008-b12 show little difference, with overall C α RMSD of 0.67 Å and 0.75 Å, respectively (Figure R1a-c). The alignment of the AMC008-VRC01-35O22 C1 structure to the C3 structure shows an overall C α RMSD of 0.88 Å, with a rigid body shift of ~2 Å of the Env base. However, the protomers of the AMC008-VRC01-35O22 C1 structure align well with the protomer of AMC008-VRC01-35O22 C3 structure, with overall C α RMSD of 0.57 Å, 0.44 Å, and 0.55 Å, respectively (Figure R1d). As shown in Supplementary Figure 6a, we combined the two and three 35O22-bound classes when processing the AMC008-VRC01-35O22 C3 map, and the difference on the C1 and C3 Env base was likely introduced here. We also reprocessed the AMC008-PGT121-VRC01 map in C3, a byproduct of the AMC008-PGT121-VRC01-35O22 dataset and built the atomic model. The AMC008-PGT121-VRC01 C3 structure aligns well with the AMC008-VRC01-35O22 C3 structure, with an overall C α RMSD of 0.49 Å, indicating the AMC008-VRC01-35O22 C3 structure, though it differs from its C1 structure, is a good representative of closed Env (Figure R1e). In summary, there are no significant differences between the three C3 structures and their C1 counterparts, including the 540-650 range and FP. The glycans of the C1 structures differ only subtly, mainly due to the resolution loss reconstructing the cryo-EM maps in C1.

Figure R1. C α RMSD of the AMC008-CD4 (a), AMC008-b12 (b), and AMC008-VRC01-35O22 (c) C1 structures in comparison to C3 structures, protomers of AMC008-VRC01-35O22 C1 structure in comparison to C3 structure (d), and AMC008-PGT121-VRC01 C3 structure in comparison to AMC008-VRC01-35O22 C3 structure (e). The PGT121, b12, VRC01, 35O22 Fabs, and CD4 are hidden for better visualization.

More analysis/discussion on potential glycan effects (or no effect) on the lower stoichiometry of 3BC315 would be helpful. Somewhat similar to what is shown for b12.

We thank the Reviewer for this suggestion. To accommodate the Reviewer's request, we re-interpreted our data: Two glycans, N88 and N625, are interacting with 3BC315 at the paratope-epitope interface. The glycan effects on 3BC315 binding are hard to interpret solely based on the structural analysis, as the reviewer has pointed out in a previous comment. The Fab:glycan interactions identified from structural analysis may not reflect actual contribution to binding. We were able to analyze the glycan effects on b12 binding because Env cannot bind b12 in the closed state due to heavy clashes of b12 with the glycans indicated; clashes that cannot be alleviated by glycan movements.

The Ward lab at Scripps has published work that investigates the effects of glycans on 3BC315 binding biochemically (PMID: 26404402). It was found that glycan N625 subtly affected 3BC315 binding affinity or neutralization potency and that deletion of glycan N88 resulted in higher binding affinity and a 30-fold increase in neutralization. These results align with our structural analysis that glycan N88 remodels greatly to accommodate 3BC315 binding while glycan N625 does not.

As glycan N88 on the more open AMC008-b12-3BC315 structure requires similar conformational rearrangements as in the case of closed Env, we infer the increased

3BC315 binding stoichiometry is driven by the conformation preference of 3BC315, not by glycan effects.

Figure 6 is just difficult to interpret. I appreciate that the authors are trying to depict a very complicated system but any effort to make this a little cleaner or more interpretable would be appreciated.

We thank the reviewer for this comment. The conformational landscape of Env is among the most complicated in nature. The present study relays structures of multiple novel Env structures and conformations. Our analysis is expanded by integrating knowledge across the literature; a vast task, that, to our knowledge, has not been provided elsewhere in the literature. Moreover, we note another Reviewer described our Fig. 6a as “It has been speculated that Env features a multi-dimensional landscape, and I have never seen this documented in a more impressive way than in Fig. 6a.” To accommodate the Reviewer, we modified the Discussion section to more clearly relay former Fig. 6 current Fig. 7 (Lines 346-373).

While speculative, I'd like to see discussion of any other commonalities in other viral fusion proteins that this might potentially apply to. This would help with the general impact by extending the conformational pathway to other viruses.

The insights from this study, together with prior studies on HIV-1 Env dynamics, were largely gained from the Envs in soluble SOSIP form, which lack the MPER and TM domains that play crucial roles in the fusion pathway. Additionally, the conformational pathway proposed in this study only covers ligand-free Env to CD4-and-coreceptor-bound Env; the field still has little knowledge of events happening after coreceptor binding.

Zoonotic origin Envs of HIV-1 (SIVcpz and SIVgor), HIV-2, and zoonotic origin Env of HIV-2 (SIVsmm) exhibit high homology to HIV-1 Env and they most likely follow a similar fusion mechanism. In fact, structures of SIVcpz (MT145K, PDB: 6OHY) and SIVmac (SIV E660.CR54, PDB: 8DUA; SIVmac239, PDB: 8DVD, 7T2P, and 7T4G) Envs have been determined and show high similarity to HIV-1 Env.

In accordance with The Reviewer's suggestion, we have added the following phrase to our discussion:

Lines 411-413: Considering the sequence homology and structural similarities of HIV-1 Env, it is likely that Envs of SIVcpz, SIVgor, HIV-2 and SIVsmm follow similar conformational trajectories⁴⁵⁻⁴⁷.

Line 540 “PCT121” should be PGT, “Rn32” should be RnS2

Typos corrected.

Reviewer #2 (Remarks to the Author):

Jiayan Cui and colleagues study the opening of the HIV-1 Env trimer by CD4 receptor for fusion. Because the HIV-1 Env is a master of immune evasion, the mechanism of how Env is opened is of great importance, not just for the understanding of these viral fusion machines, but also for immunogen design to generate vaccines that can neutralize this virus. To gain structural insights into the mechanism of Env opening, the authors engineered a stabilized SOSIP.v4.2 trimer based on the HIV-1 strain, AMC008, which appears somewhat resistant to opening. Using this Env and a panel of neutralizing antibodies along with soluble CD4 receptor, the authors determined several Env structural intermediates and asymmetric states during the transition from closed to open Env.

For instance, in complex with the VRC01 and 35O22 Fabs, the authors determine the structure of a closed Env (designated AAA). Using the Fab 3BC315 (binding gp41-gp41 interface), the authors next identify a base-relaxed AAABR Env configuration. Using the CD4 binding site targeting antibody b12, the authors identify a partially open (more closed than B41-b12, hence designated “moderately open” Env, BBB), and with CD4 a moderately open configuration (B’B’B’). These structures are compared to partially open CCC and fully open DDD, which is the most open Env configuration that likely moves closest towards the membrane to engage the coreceptor. In summary, the study identifies novel conformational states of Env, from the fully closed, a “base-relaxed” state, a “moderately open” state, to partially open, then finally fully open, as well as asymmetric configurations thus revealing a complex conformational landscape for the dynamic opening of HIV-1 Env. The report is incredibly comprehensive and I have only minor suggestions that the authors could consider.

We thank the reviewer for their kind evaluation of our work and the recommendation for publication. Our point-by-point comments addressing the issues appear below.

Looking at the final model for the HIV-1 Env opening presented in the Fig. 6a scheme, would it be fair to mention in the discussion that this is consistent with a multidimensional conformational landscape with parallel paths where each HIV-1 isolate, and each Env mutant might follow a slightly different path? It has been speculated that Env features a multi-dimensional landscape, and I have never seen this documented in a more impressive way than in Fig. 6a.

We thank the Reviewer for their positive evaluation of former Fig. 6, current Fig. 7. Authors agree that it is highly likely that the conformational pathway of Env, transitioning from a closed to a fully open state, is multidimensional and varies across Env isolates and environments. We have rephrased our discussion accordingly.

Lines 359-373: The equilibrium among these conformations varies by viral isolate: tier 2 and 3 isolates are strongly biased towards the closed state, whereas tier 1 isolates populate open states more frequently. The effects of CD4 engagement further illustrate this isolate dependence. CD4 binding proceeds in a stepwise manner, with the

response to the first CD4 molecule differing markedly across isolates. For BG505, a tier 2 Env, binding of a single CD4 yields a mixture of conformations, with approximately two-thirds of trimers remaining closed and one-third becoming asymmetrically open. In contrast, BaL, a tier 1b Env, becomes fully asymmetrically open upon binding just one CD4^{13,14}. Furthermore, structural studies of antibody-bound states reinforce this theme. For example, T/F100 Env bound to 8ANC195 adopts a base-relaxed state A_{BR}, whereas the same antibody binds BG505 in a closed state A³⁰. Our AMC008 bound to 3BC315 adopts an A_{BR} state, while BG505.MD39 bound to 3BC315 remains closed⁴³. b12-bound B41 Env exhibits a fully open state D while b12-bound AMC008 exhibits the moderately open state B¹⁰. Altogether, these structural and biophysical data highlight the fact that the Env conformational landscape is both isolate-dependent and mutation-dependent, making it a truly multi-dimensional conformational landscape.

The authors present several Env structures bound to b12, representing different opening states from moderately open to partially open and fully open. As expected, gp120 models in these structures appear almost identical since b12 Fabs bind to the same epitope across all states. Furthermore, the HR1C helices of gp41 align well across these moderately, partially, and fully open structures (Fig. 2i), suggesting that changes in the gp41 bases and gp41-gp120 interfaces play a greater role in the overall opening of Env. This is likely important for the CD4 engagement on membrane-bound Env and may involve tilting of the trimer. However, the authors places heavy emphasis on HR1C direction (Fig. 2e-h and Fig. 4e), which could mislead readers into thinking HR1C movement drives Env opening. Maybe the authors could re-analyze the differences in gp41 base by superimposing the structures on HR1C and focusing the discussion on base rearrangements? It may also help to explain the variable stoichiometry of 3BC315 binding observed across different complexes.

We thank the Reviewer for this suggestion. There are three rearrangement units in gp41: (1) FP, FPPR and HR1N, which remodel greatly among conformational states; (2) HR1C assembly and (3) C-C loop and HR2, which move as rigid bodies. The (2) HR1C assembly only becomes more compact during the transition from states A/A' to states B/B', the compactness of the HR1C assembly remains the same in all B/B'/C/C'/D/D' states. The (3) C-C loop and HR2 unit move as a rigid body during the transition from states B/B' to states C/C' and from states C/C' to state D/D'. Thus, the alignments of protomers show the movement of HR1C from states B/B' to states C/C' and from states C/C' to state D/D' actually reflects the relative movement of C-C loop and HR2. The benefits of aligning C-C loop and HR2 rather than HR1C are the clarity of superimpositions and quicker identification of protomeric states.

To accommodate the Reviewer's suggestion, we added a detailed explanation to avoid misunderstanding.

Lines 156-162: Subsequent alignment of gp41 across these structures reveals distinct HR1C conformations: in partially and fully open states, the HR1C helix extends by at least one additional N-terminal turn and rotates in **divergent multiple** directions relative to the closed state. While AMC008-b12 also exhibits N-terminal helical extension of

HR1C, its orientation remains similar to a closed state (Fig. 3e). The apparent movement of HR1C arises from rigid-body shifts relative to the C-C loop/HR2 module, which is used for alignment, and not from internal rearrangements within HR1C itself.

The authors determined the structure of AMC008.SOSIP.664-b12-3BC315 complex and observed an interaction between b12 and 3BC315 Fabs, which facilitates increased accessibility of 3BC315. This was further supported by mass photometry, which revealed increased numbers of bound 3BC315. A synergistic neutralization effect was observed on b12-sensitive strains. However, such an effect was minimal on b12-resistant strains, where the IC₅₀ of 3BC315 remained similar in the presence or absence of b12. These observations suggest that b12-bound state favors 3BC315 binding, but the reverse may not be true. However, the authors interpret this interaction as cooperative binding. The authors could perform flow cytometry with sequential antibody addition (e.g., adding b12 followed by 3BC315, and vice versa) to assess whether each antibody enhances the binding efficiency of the other. Yet, again, this is not central to the main message of this manuscript.

As pointed out by the Reviewer, we demonstrate by structural analysis and mass photometry that b12 binding to Env increases subsequent 3BC315 binding. A matching synergistic neutralization effect is observed in b12-sensitive strains.

Additionally, we observed a synergistic neutralization effect of the combination of b12 and 3BC315 against b12-resistant strains. To Accommodate the Reviewer's suggestion, we have performed three additional repeats of the neutralization assay, and the results for the b12-resistant strains are shown in the figure below. While the effect on resistant strains is less pronounced than on sensitive ones, we believe that the replicates consistently show a real, albeit slight, increase in neutralization.

Figure R2. Neutralization by b12 or 3BC315 alone at predetermined concentrations and the combination of b12 and 3BC315 against b12-resistant HIV-1 pseudoviruses.

Although the structural interpretations are well supported, some figures could be clearer. For instance, in Fig. 1d, a more zoom-out view could help to show the overall context of the formed cavity; Fig. 1f, presenting the structures in separate groups may better to highlight the differences; Fig. 2i, indicating the superposition on HR1C would improve clarity; Figs. 3c, 4b, 4e, additional labels or annotations would make these figures more accessible to a broader audiences, etc. For the key model Fig. 6, would it be helpful to have a side view row to illustrate the mechanism of opening?

We thank the Reviewer for their suggestions to improve the clarity of the figures. We edited former Fig. 1-4 accordingly.

For former Fig. 6, current Fig. 7, we did prepare a version with side views; however, we found that the addition of side views makes the figure extremely verbose and significantly increases its size, without providing much additional information about the Env conformational states, given the limited size of the panels. The primary purpose of this figure is to propose a model of progressive rearrangements of HIV-1 Env during its opening and CD4 engagement. For the structural details, we present Supplementary Fig. 4, which illustrates the openness of each conformational state, and Fig. 4f, which summarizes the features of each conformational state.

The literature review is comprehensive, and then some readers may wonder why some papers were not discussed. For instance, could a recent paper by Bhishem Thakur et al., and Priyamvada Acharya (PMID: 40382314) be discussed? Also, Youdong Mao who's reputation had been damaged from his early structural work on HIV-1 Env, has however solved recent structures that follow a mainstream structural approach as well as mainstream thinking of Env opening, includes insights into asymmetric states and the MPER peptide (PMIDs: 37202420, 40089599).

We thank the Reviewer for these suggestions. The recently published work from the Priyamvada Acharya lab describes three alternative Env conformational states in the context of fusion peptide directed antibody (VRC34.01) bound. The consequence of VRC34.01 binding is dramatic redirection of the fusion peptide (as exemplified in former Fig. 1f, current Fig. 2f), similar to VRC34.01-bound closed Env (PDB: 5I8H). We believe that the binding of VRC34.01 decouples the fusion peptide conformation from each conformational state, turning the fusion peptide conformation exclusively outward-directed from the Env core. Thus, these three VRC34.01-induced Env states do not align with the fusion peptide preference of our model (former Figs. 3f and 6, current Figs. 4f and 7). If the fusion peptide conformation in the three conformational states is disregarded, they would classify as states B'B'C', B'C'C', and C'C'C', respectively. As the three classified states align with our model disregarding fusion peptide positioning, and the positioning of the fusion peptide is a consequence of VRC34.01 binding, we did not incorporate this work into our analysis but we have included these three conformational states in Extended Data Table 1 of our initial submission (current

Supplementary Table 1). To accommodate the Reviewer's suggestion and since these structures do reflect the conformational plasticity of the fusion peptide, and inclusion is also recommended by another Reviewer, we added a discussion of the study to make our model of Env opening even more rigorous:

Lines 405-411: A recent study reported three CD4-bound Env structures stabilized by the FP-binding antibody VRC34.01⁴⁴. In these structures, VRC34.01 binding reorients the FP outward, decoupling its conformation from the states in our proposed model. If classified without considering FP conformation, these structures correspond to the B'B'C', B'C'C', and C'C'C' states (Supplementary Fig. 4q-s and Table 1). These new states effectively bridge the gap between the moderately open (B/B') and partially open (C/C') states in our present model.

The two articles from the Youdong Mao lab describe four Env structures. These structures were solved in the context of full-length Env, though MPER and TM regions were largely disordered. These Env structures are all in closed conformation with more ordered and extended HR1N in two protomers of the Env. The authors infer that the heterogeneity and asymmetry in HR1Ns of these Envs are likely caused by the Env tilting on the membrane and the state of MPER. As our AMC008 Env structure ensemble and the Env structures we reference are all in SOSIP form and do not have MPER and TM regions, we do not include these structures in our model. Nevertheless, the two studies demonstrate the highly flexible nature of gp41 in closed Env.

Additionally, and in line with the Reviewer's request, we summarized all the non-closed Env structures to date in former Extended Data Table 1, current Supplementary Table 1.

1. Line 234: for consistency, use "AMC008-b12-3BC315" throughout.
2. Line 303: it should be "Fig. 5e,f"

Typos corrected.

Reviewer #3 (Remarks to the Author):

The manuscript titled “Conformational Landscape of HIV-1 Env from Closed to Fully Open” by Cui et al., determines multiple new structures of the HIV-1 Env defining conformational states in the context of gp120/gp41 targeting antibody 3BC315. The paper provides several new structures of different HIV-1 Env conformations, both in the closed and partially open states. Additionally, they provide structure-function insights into the synergy between antibodies b12 and 3BC315. The structural determinations are well done providing atomic level details of interaction interfaces and local conformations. The mass photometry and neutralization experiments showing synergy with b12 and 3BC315 are nicely paired with the structural analysis. HIV-1 Env conformations have been extensively studied over many years. This study adds valuable information to the vast knowledge that has already accumulated on this topic.

We thank the reviewer for their kind evaluation of our work.

The authors should address the following:

1. Newer literature on the topic, especially studies that are highly relevant to the results and conclusions on this paper, should be better integrated in the analysis. Recently, a study was published (PMID: 40382314) that reported on the “Conformational trajectory of the HIV-1 fusion peptide during CD4-induced envelope opening”. The authors have included three structures from the study in Extended Data Table 1 but should also integrate within the analysis in the main text (and cite the published study). Some places where the suggested integration can happen:
 - a. Line 133-134: The authors state that: “Taken together, these results underscore FP conformational plasticity across closed, partially open, and fully open Env states.” The conclusions in the published study mentioned above are highly synergistic with this stated conclusion, reaching essentially the same conclusion with a different gp120/gp41 interface binding antibody (the FP directed VRC34.01). This would be a great place to refer to the published study and align the conclusions.
 - b. The structures (at least 9D90, 9D98 and 9D8Y) should be included in Extended Data 3 (and described in the text along with the comparisons with the other published structures).
 - c. In the section titled “CD4-bound moderately open AMC008” (line 177) authors should include comparison with CD4-bound structures: 9D8Y, 9D98 (PMID: 40382314). Also, include these structures in the comparisons described in lines 234-238.
 - d. Figure 3f: the authors indicate State C’ as having a buried FP helix based on the 6CM3 structure, but this description does not consider new data that shows that this state geometry can accommodate the FP in both a buried configuration (as in the 8ANC195-bound 6CM3 structure) and in an accessible/exposed configuration (as in the VRC34.01-bound 9D90 structure). By not incorporating the newer data into their analysis, nuances that have enriched our knowledge of these states are being lost. Authors should revise.

We thank the Reviewer for these suggestions. The recently published work from the Priyamvada Acharya lab describes three alternative Env conformational states in the context of fusion peptide directed antibody (VRC34.01) bound. The consequence of VRC34.01 binding is dramatic redirection of the fusion peptide (as exemplified in former Fig. 1f, current Fig. 2f), similar to VRC34.01-bound closed Env (PDB: 5I8H). We believe that the binding of VRC34.01 decouples the fusion peptide conformation from each conformational state, turning the fusion peptide conformation exclusively outward-directed from the Env core. Thus, these three VRC34.01-induced Env states do not align with the fusion peptide preference of our model (former Figs. 3f and 6, current Figs. 4f and 7). If the fusion peptide conformation in the three conformational states is disregarded, they would classify as states B'B'C', B'C'C', and C'C'C', respectively. As the three classified states align with our model disregarding fusion peptide positioning, and the positioning of the fusion peptide is a consequence of VRC34.01 binding, we did not incorporate this work into our analysis and only included these three conformational states in Extended Data Table 1 of our initial submission (current Supplementary Table 1). Since these structures do reflect the conformational plasticity of the fusion peptide, and inclusion is also recommended by another Reviewer, we added a discussion of the study to our Discussion section to make our model of Env opening even more rigorous:

Lines 415-421: A recent study reported three CD4-bound Env structures stabilized by the FP-binding antibody VRC34.01⁴⁴. In these structures, VRC34.01 binding reorients the FP outward, decoupling its conformation from the states in our proposed model. If classified without considering FP conformation, these structures correspond to the B'B'C', B'C'C', and C'C'C' states (Supplementary Fig. 4q-s and Table 1). These new states effectively bridge the gap between the moderately open (B/B') and partially open (C/C') states in our present model.

We included the structures (9D90, 9D98, and 9D8Y) in former Extended Data Fig. 3, current Supplementary Fig. 4.

5. line 49-50: "CD4 engagement begins with the binding of one CD4 molecule, resulting in an asymmetric open or closed conformational state of Env." This sentence is confusing and should be rephrased. I think what the authors mean here is that CD4 can bind to both closed and open Env conformations. The "result" of CD4 binding, however, is Env opening.

We thank the Reviewer for this comment. The ultimate result of CD4 binding is Env opening and exposure of the coreceptor binding site. However, the engagement of multiple CD4 molecules to Env is a stepwise process, and the responsiveness of Env to CD4 varies across HIV-1 isolates and tiers. The initial CD4 engagement of binding of one CD4 molecule to BG505 Env, a tier 2 HIV-1 isolate, results in two-thirds of Env trimers adopting the closed state, and one-third of Env trimers adopting the asymmetric open state (PMID: 37993719); in contrast, binding of one CD4 molecule to BaL Env, a tier 1b HIV-1 isolate, results in all Env trimers adopting the asymmetric open state (PMID: 37993716). The subsequent CD4-binding (a second or an additional two copies of CD4) to BG505 or BaL Env results in asymmetric or symmetric opening of Env

(PMIDs: 37993716, 37993719).

We clarified our sentence as follows:

Lines 49-50: CD4 engagement begins with the binding of one CD4 molecule, resulting in either a closed or an asymmetric open ~~or-closed~~ conformational state of Env.

6. Lines 53-54: “Notably, cryo-ET reveals that three CD4-bound Env adopts a partially open state distinct from the fully open state typically observed by cryo-EM^{11,14}”. This sentence is only partially correct. While the cited paper (PMID: 37993716), captures a state that is partially open, earlier cryo-ET studies (PMID: 18668044), albeit at lower resolutions, described open structures that aligned well with the open structure of B41 Env bound to CD4 and 17b resolved by single particle cryo-EM analysis (see Extended Figure 3a in PMID: 28700571). Differences in Env used or the specific membrane context used in the studies could be responsible for the differences. Authors should revise the sentence to be better reflective of the published data in the literature.

We thank the Reviewer for this comment. We emphasize that the CD4-bound fully open Env captured by the earlier study (PMID: 18668044) is in the context of Env complexed by soluble CD4 and 17b, while the partially open CD4-bound Env captured by the recent and cited article is in the context of Env complexed by membrane-attached CD4. In addition, another earlier cryo-ET study captured CD4-bound Env in the context of soluble CD4 and 17b; the gp41 conformation does not resemble either a partially open or fully open structure, but rather resembles two asymmetrically open BG505-E51-sCD4 structures, which we classify as C'C'D' and C'D'D' (PMID: 32601441). In this same study, the authors observe that gp120 would not shed when incubated with MLV-CD4, but would shed when incubated with soluble CD4 and 17b, aligning with a previous study that shows membrane-attached CD4-bound Env is longer-lived than soluble CD4-bound Env (PMID: 19343205).

In conclusion, the effects of soluble CD4 binding and membrane-attached CD4 binding to Env are different; the soluble CD4-bound Env is more open and shorter-lived than membrane-attached CD4-bound Env. While we agree that differences in Env strain and context will likely affect the conformation of CD4-bound Env, the partially open Env state reported in the recent and cited article is the far more biologically relevant representative of the three CD4-bound Env on a cell surface (PMID: 37993716). We infer that the fully open state, as captured by soluble CD4-bound Env in both cryo-EM/cryo-ET studies, is a result of the loss of membrane tension, and the fully open Env state is possibly the state that the coreceptor binds (also see response for comment 15).

To clarify and accommodate the Reviewer's request, we edited our writing as follows:

Lines 52-54: Notably, cryo-ET reveals that **binding of three membrane-attached CD4 molecules results in Env adopting** a partially open state distinct from the fully open state typically observed by cryo-EM^{11,14}.

7. In line 64 and 67, 3BC315 is described as “targeting the gp41-gp41 interface” In line 96-97, its described as “3BC315 binds predominantly to one gp41 subunit but exhibits additional interactions with gp120 (same protomer) and adjacent gp41.” Line 106: “3BC315 CDRH3 loop inserts into a cavity at the gp120-gp41-gp41 interface” Given that 3BC315 contacts both gp120 and gp41 subunits, it may be better to call it a gp120-gp41 interface targeting antibody. In any case, authors should examine these diverse descriptions in the manuscript and try to homogenize.

We thank the Reviewer for pointing out this apparent inconsistency. We followed the epitope description found in the literature in our introduction which stems from the Ward Lab (PMID: 26404402). Here, 3BC315 was characterized as a gp41-gp41 interface binder. Later in our manuscript and based on our atomic-resolution structures, we re-characterize 3BC315 as a gp120-gp41-gp41 interface antibody. Following the Reviewer’s recommendation, we edited the sentences as follows:

Lines 64-69: We identified novel Env conformations by complexing AMC008 with antibody 3BC315 (targeting the gp120-gp41-gp41 interface)²³, antibody b12 (targeting the CD4 binding site; CD4bs)²⁴, CD4, or a combination of 3BC315 and b12. 3BC315 and b12 were chosen based on prior insights: b12 facilitated the first cryo-EM structure of occluded open Env¹⁰, while 3BC315 destabilizes Env via its interactions with the interprotomer interface²⁵.

8. Figure callouts are out of sequence. For example, Extended Data Fig. 5d is called out first, then Extended Data Fig. 2d, then figure 1c and so on. This makes it difficult for the reader having to go back and forth through the figures. The figures and callouts should be re-organized to go in sequence.

We thank the Reviewer for bringing this to our attention. We moved Extended Data Fig. 1 to Figure 1, reorganized the panels of Figure 1, Extended Data Fig. 1, and Extended Data Fig. 2 to align with the text, and edited the figure legends accordingly. We reorganized the panels of Extended Data Figs. 5 and 6 (cryo-EM data processing schemes) to align with the sequence of the structures being discussed in the text and edited the figure legends accordingly. All other figures and legends are edited accordingly.

9. The section titled: “3BC315-bound base-relaxed AMC008” begins with the conclusion “The AMC008-PGT121-VRC01-3BC315 structure reveals remodeling of the Env base as compared to occluded closed Envs9.” It would be better instead if the results were presented first, followed by the conclusion.

We thank the Reviewer for this comment. We revised the paragraph accordingly:

Lines 110-112: To enable comparison of our 3BC315-bound AMC008 Env structures with closed AMC008 Env, we solved a 2.9 Å cryo-EM structure of closed AMC008 by

complexing AMC008 with VRC01 Fab and 35O22 Fab (targeting the gp120-gp41 interface) (Fig. 2a)²⁸.

The AMC008-PGT121-VRC01-3BC315 structure reveals remodeling of the Env base as compared to occluded closed Envs⁹. To investigate this base remodeling, we solved a 2.9 Å cryo-EM structure of occluded closed AMC008 by complexing AMC008 with VRC01 Fab and 35O22 Fab (targeting the gp120/gp41 interface) (Extended Data Fig. 2a,6a)²⁸.

10. Lines 113-115: “Superimposition of the AMC008-PGT121-VRC01-3BC315 and occluded closed AMC008-VRC01-35O22 structures demonstrates dramatic displacements by more than 5 Å in epitope-associated structural elements upon 3BC315 binding (Fig. 1b).” Here the authors compare the 3BC315 bound structure with the 35O22-bound structure to draw conclusions about the structural rearrangements caused by 3BC315. Since, 35O22 is itself a gp120/gp41 interface targeting antibody, it would be helpful to add a comparison of 35O22 (perhaps in the Supplement) with a closed Env structure that does not have an antibody bound at the gp120/gp41 interface.

We thank the reviewer for this suggestion. We reprocessed the AMC008-PGT121-VRC01 map, a byproduct of the AMC008-PGT121-VRC01-35O22 dataset and built the atomic model. The AMC008-PGT121-VRC01 structure aligns well with the AMC008-VRC01-35O22 structure, with an overall C α RMSD of 0.49 Å, confirming the binding of 35O22 does not significantly alter the gp120/gp41 interface compared to the remodeling of the gp120/gp41/gp41 interface upon 3BC315 binding.

Figure R3. C α RMSD of the AMC008-PGT121-VRC01 structure in comparison to the AMC008-VRC01-35O22 structure. The PGT121, VRC01, and 35O22 Fabs are hidden for better visualization.

12. Lines 157-159: “Integrating gp120 openness, gp41 conformation, and CD4-binding status, we categorized Env states as follows: A (closed), ABR (base-relaxed closed), B (moderately open), C (partially open), C’ (CD4-bound partially open), D (open) and D’ (CD4-bound open).” This sentence occurs somewhat prematurely without adequate details about the rationale behind the nomenclature. Those details come later in the manuscript (lines 204-217), so the authors should consider some re-organization/clarifications.

On the same note, many of these states have been described before in the literature and were named a certain way. It would be better to either adhere to the nomenclature

that these states are known by in published literature or, when warranted, to provide clearly stated rationale for renaming them. For example, in lines 126-127, authors states: “In contrast, in occluded fully open B41-b12 and CD4-bound accessible fully open B41-17b-CD4”, calling both the b12-bound B41 structure and the CD4/17b-bound B41 structure “fully open” is confusing. The b12-bound B41 Env structure was previously described as “open occluded” (PMID: 35136084).

To summarize, it is highly recommended that authors examine their system of nomenclature, align it with published literature, changing and redefining only when necessary.

We thank the Reviewer for these observations. In our classification system, there are three criteria: Presence of CD4, openness of gp120, and conformation of gp41. The A_{BR} state is a variant of the closed state. Thus, at the manuscript stage of lines 157-159, it is suitable to introduce our nomenclature system, as we have compared the gp120 openness and gp41 conformation across closed and open conformational states with or without CD4 binding. The benefit of introducing the nomenclature system at an earlier stage of the Results section is that subsequent discussions have clear indications for each state. Moreover, the content that follows provides further details about these conformational states. The content of lines 204-217 is a further summary of our classification and nomenclature system. The next section expands our classification and nomenclature system to asymmetric Envs.

When open HIV-1 Env structures (B41-b12 and B41-17b-CD4) were first described by Ozorowski, Pallesen *et al.* in atomic detail, the authors did not distinguish between these two structures in their naming, although they analyzed the differences between them (PMID: 28700571). Later, the Bjorkman lab reported two other open HIV-1 Env structures captured by antibodies other than b12; they thus described these antibody-bound open structures as occluded open, as the coreceptor binding site on these structures is occluded by the V2 loop, different from CD4-bound open structures, where the coreceptor binding site is accessible (PMID: 35136084). The Bjorkman lab also reported a less open CD4-bound BG505-8ANC195-17b-CD4 structure, describing it as partially open compared to the B41-b12 structure (PMID: 30308160). Since then, all antibody-bound open structures have been described as occluded open; partially open describes the isolated example of BG505-8ANC195-17b-CD4 cryo-EM structure (also observed in cryo-ET studies); and all asymmetric open structures have been generalized as asymmetric open.

This nomenclature system has mainly three flaws: (1) the coreceptor binding site of CD4-bound Env or CD4-bound gp120/gp41 protomer is also occluded when the Env is in closed (PMID: 37993716, 28218750) or moderately open state (present study); (2) the partially open state could also be non-CD4-bound and occluded; (3) the diverse asymmetric open conformational states could not be distinguished.

We agree that the inconsistency and shortcoming of naming in the literature is confusing. Thus, we name these states three-dimensionally by Env openness,

coreceptor binding site accessibility, and CD4 binding. However, accommodating the Reviewer's recommendation, we adapted our nomenclature system to align with existing literature while maintaining accurate description of the conformational states. The current nomenclature system is as follows: A (closed), A_{BR} (base-relaxed closed), B (occluded moderately open), B' (CD4-bound occluded moderately open), C (occluded partially open), C' (CD4-bound partially open), D (occluded fully open), and D' (CD4-bound fully open). The disambiguation of CD4-bound and occluded is necessary, as they were not completely opposed, and the prefixation of "fully" to state D/D' is also instrumental, as open is too vague. The asymmetric open states are named by their protomeric state with three letters.

13. There is opportunity for improving the analysis of the b12-bound states. In the study, the authors identified a moderately open b12-bound Env and have compared it to the the b12-bound B41 Env structure (more open). Authors should also include in their analysis the structure of b12 bound to an engineered closed Env (PMID: 38306419; Figure 5 E, F). Additionally, they should also refer to the analysis in PMID: 28700571, Extended Figure 5F. All these structures, published and the ones determined in this study, taken together, show that b12 can bind to Envs in different conformations, although a more open conformation is preferred.

We thank the Reviewer for this suggestion. The Env used to form the complex of b12-bound closed Env is heavily engineered (PMID: 38306419). There are two modifications that we believe would affect the b12-bound Env state: the apex-staple interprotomer disulfide bond inherently prevents the opening of Env, and the N197D glycan deletion. These modifications sterically allow b12 binding to this closed, engineered Env. The authors themselves pointed out in the study, that the ability of b12 binding to the apex-staple Env is likely achieved by N197 glycan elimination. In our analysis, the N197 glycan is reoriented from an outward to an inward orientation relative to the Env threefold symmetry axis to sterically accommodate the binding of b12. This reorientation brings the N197 glycan closer to the V2/V3 loops of the adjacent protomer. In the closed conformational state, the reoriented N197 glycan would severely clash with the V2/V3 loops of the adjacent protomer (former Fig. 2c,d, current Fig. 3c,d). That is, the conformational change of N197 upon b12 binding prevents the retention of Env in the closed state. As the N197 glycosylation site is highly conserved across HIV-1 strains (PMID: 23384254), we consider the b12-bound closed state as a special engineered case.

14. Line 240-241: "Despite these similarities, gp120s of AMC008-b12-3BC315 lack the features of a CD4-bound accessible open conformation (Fig. 4d)." – authors should state here what these features are that differentiate the states.

We thank the Reviewer for this suggestion. We edited the sentence as follows:

Lines 248-251: Despite these similarities, gp120s of AMC008-b12-3BC315 lack the features of a CD4-bound ~~accessible~~-open conformation, which are displaced V1V2 loops, accessible coreceptor binding site, and formation of α 0 helix and 4-stranded

bridging sheet.

15. It is unclear what Figure 4g is showing that is new and derived from the results of this study. These results and conclusions were described in previous studies where these structures were published, in particular, in PMID: 37993716. This section should be re-organized to perhaps include in the discussion instead of stating it as a result of this study. The EMDB # of the cryo-ET map should be included in the figure legend.

We thank the Reviewer for bringing this to our attention. The cited cryo-ET study (PMID: 37993716) reveals that membrane-attached CD4-bound Env is partially open, and the authors also noted that in this partially open state, the membrane distance between the target cell and viral membrane is too long to engage the coreceptor. The authors hypothesize that the shortening of the membrane distance, which allows the engagement of the coreceptor, is achieved by either release of CD4 or bending of the target cell membrane (PMID: 37993716, Fig. 4l).

In former Fig. 4g, current Fig. 5g and elsewhere, and different from the analysis in PMID: 37993716, we argue that the conformational change of CD4-bound Env from partially open state C' to fully open state D' causes a shortening of the distance between the target cell membrane and viral membrane (by ~30 Å). This shortening of the membrane distance is adequate to engage the coreceptor. Our analysis endows the biological relevance to state D', namely that state D' is primed for coreceptor engagement, which has not yet been captured by cryo-ET (in the context of full-length, membrane embedded CD4).

To accommodate the Reviewer's comment, we have added PDB and EMDB numbers of the models and the cryo-ET density map used for our analysis in the figure legend.

16. Figure 2c, legend: "b12 cannot bind AMC008 in closed conformation due to clashes of N197 and N301 glycans with b12. Clashes are indicated by red * marks. d Binding of b12 requires minimum openness of AMC008 gp120 to accommodate N197 and N301 glycans." Authors should rephrase this since modeled clashes with glycans do not necessarily preclude binding as glycans can move to accommodate binding (as the authors have themselves shown).

We thank the Reviewer for sharing their view. Authors have not in the present study or elsewhere shown that N197 and N301 glycans can move in a closed-conformation Env to accommodate b12 binding by relieving steric clashes. In fact, our analysis indicates that the conformational change necessary for N197 to accommodate b12 binding prevents the retention of Env in the closed state (former Fig. 2c,d, current Fig. 3c,d, and see response to comment 13).

17. Figures showing local resolution of the cryo-EM maps should be included in the Supplement.

We thank the Reviewer for this suggestion. We processed local resolution estimations for all the cryo-EM maps, based on which we built the models. We added Supplementary Fig. 6 to present the local resolution of these cryo-EM maps.

Supplementary Fig. 6 | Local resolution estimation for the cryo-EM density maps of AMC008 in complex with Fabs and CD4.

Reviewer #1 (Remarks to the Author):

The authors have addressed my major concerns. I recommend acceptance but provide the following comments that I hope will be taken into account to help improve the final manuscript.

We thank the Reviewer for their valuable suggestions to improve our manuscript and for their recommendation to publish our work.

I noted this in my initial summary but maybe should have included it more specifically as an enumerated issue to address. The authors make sweeping assumptions throughout the manuscript about antibody-bound, stabilized states on soluble Env representing natural intermediates of Env that are membrane-bound and lack stabilization. I'm even on their side for the most part but the readers need to be better informed of the assumptions and limitations presented here compared to membrane-bound Env on a virion. This could be with caveats in the introduction and the discussion, but it should be thorough.

We thank the Reviewer for this suggestion. To accommodate this, we added caveats in the introduction and discussion:

Lines 68-71: Integrating these results with previous studies on soluble Env, we propose a classification framework to characterize Env conformations and a mechanism by which HIV-1 Env transitions from closed to CD4-bound fully open en route to membrane fusion.

Lines 379-384: It is important to note that this model is derived from studies of soluble, stabilized Env constructs. On the native membrane, the energy landscape and transition kinetics may differ due to the absence of the transmembrane domain and cytoplasmic tail, and the presence of artificial stabilizations. Nevertheless, our present model provides a structural framework for understanding the conformational plasticity of HIV-1 Env and serves as a valuable template for therapeutic development and immunogen design.

Authors have trimmed down the relevance of the incidental direct interaction of the two antibodies in the results but have left it in the abstract. While it's technically correct, authors may want to focus on the more relevant conformational synergistic results from binding the two antibodies.

We thank the Reviewer for this suggestion. We rephrased the abstract and introduction to narrate the experiments we conducted rather than highlighting the interactions between the two antibodies.

Lines 21-23: Observing enhanced 3BC315 binding occupancy in the presence of b12, we investigate the cooperativity of these antibodies using mass photometry and neutralization assays.

These combination studies allowed us to capture previously uncharacterized HIV Env conformational states. We observed enhanced 3BC315 binding occupancy in the presence of b12 and discovered that when engaging Env, antibodies 3BC315 and b12 interact with each other directly. Moreover, we decipher the allosteric mechanisms of

Env, resulting in the cooperative accommodation of 3BC315 and b12, which leads to higher occupancy and increased neutralization potency.

Lines 71-72: Lastly, we demonstrate increased cooperativity between 3BC315 and b12, two neutralizing antibodies from different human subjects, in terms of affinity and neutralization.

We observed enhanced 3BC315 binding occupancy in the presence of b12, and discovered that when engaging Env, antibodies 3BC315 and b12 interact with each other directly. Moreover, we decipher Env allosteric mechanisms underlying the cooperative accommodation of 3BC315 and b12, resulting in higher occupancy and increased neutralization.

Reviewer #2 (Remarks to the Author):

The reviewers have addressed our previous concerns and we support publication!

We are pleased the Reviewer supports publication of our work.

Reviewer #3 (Remarks to the Author):

The authors have thoroughly addressed reviewer comments. I have no further questions.

We are pleased the Reviewer supports publication of our work.